# Antimicrobial resistance and population genomics of multidrug-resistant *Escherichia coli* in pig farms in mainland China

Zhong Peng [1,2,5], Zizhe Hu[1,2,5], Zugang Li[1,2], Xiaoxue Zhang[1,2], Chaoying Jia[1,2], Tianzhi Li[1,2], Menghong Dai[3], Chen Tan[1,2], Zhuofei Xu[4], Bin Wu [1,2], Huanchun Chen[1,2] & Xiangru Wang [1,2✉]

The expanding use of antimicrobials in livestock is an important contributor to the worldwide rapid increase in antimicrobial resistance (AMR). However, large-scale studies on AMR in livestock remain scarce. Here, we report findings from surveillance of *E. coli* AMR in pig farms in China in 2018–2019. We isolated *E. coli* in 1,871 samples from pigs and their breeding environments, and found AMR in *E. coli* in all provinces in mainland China. We detected multidrug-resistance in 91% isolates and found resistance to last-resort drugs including colistin, carbapenems and tigecycline. We also identified a heterogeneous group of O-serogroups and sequence types among the multidrug-resistant isolates. These isolates harbored multiple resistance genes, virulence factor-encoding genes, and putative plasmids. Our data will help to understand the current AMR profiles of pigs and provide a reference for AMR control policy formulation for livestock in China.

[1] State Key Laboratory of Agricultural Microbiology, College of Veterinary Medicine, Huazhong Agricultural University, 430070 Wuhan, China. [2] Key Laboratory of Preventive Veterinary Medicine in Hubei Province, Cooperative Innovation Centre for Sustainable Pig Production, 430070 Wuhan, China. [3] MOA Key Laboratory of Food Safety Evaluation/National Reference Laboratory of Veterinary Drug Residue (HZAU), Huazhong Agricultural University, 430070 Wuhan, China. [4] Shanghai MasScience Biotechnology Institute, Shanghai, China. [5] These authors contributed equally: Zhong Peng, Zizhe Hu. ✉email: wangxr228@mail.hzau.edu.cn

Antimicrobial resistance (AMR) is one of the most urgent threats to global public health. Recently, the emergence of and rapid increases in multidrug-resistant (MDR), extensively drug-resistant (XDR), and even pandrug-resistant (PDR) bacteria, particularly bacteria resistant to last-resort drugs (carbapenems, colistin, and tigecycline), have catalyzed serious concern[1–4]. Several drug-resistant bacteria are of great concern worldwide, among which *Escherichia coli* is particularly notable[1,5,6]. This bacterial species is not only a leading cause of foodborne infections but also represents a major reservoir of antimicrobial resistance genes (ARGs) due to its high capacity to accumulate ARGs, mostly through horizontal gene transfer[7]. A recent study showed that the total economic cost of AMR in *Staphylococcus aureus*, *E. coli*, *Klebsiella pneumoniae*, *Acinetobacter baumannii*, and *Pseudomonas aeruginosa* has reached $2.8 billion per year in the United States[8]. In addition, *E. coli* has been found to play important roles in the dissemination of $bla_{NDM-1}$, *mcr*, and/or *tet*(X3)/(X4); these ARGs mediate resistance to last-resort drugs (carbapenems, colistin, and tigecycline) in gram-negative bacteria, which may lead to unavailability of antimicrobials in both human and veterinary medicine[3,4,9]. In addition, extended-spectrum beta-lactamase (ESBL)-producing *E. coli* is also a major global health problem[1,5]. Therefore, *E. coli* has been commonly used as a biomarker to monitor AMR on livestock farms and/or in hospitals[10–12].

The widespread and expanding use of antimicrobials in food animals is considered one of the main reasons for the worldwide rapid increase in AMR[13]. In the past two decades, meat production has plateaued in high-income countries but has greatly increased in low- and middle-income countries, particularly in Asia[14]. Against this background, antimicrobials are widely used in livestock for growth promotion and/or health maintenance. Globally, 73% of all antimicrobials sold are used in animals raised for food[14]. The widespread and expanding use of antimicrobials can lead to the development of drug-resistant bacteria in animal guts, and these resistant bacteria can spread to humans[1]. To date, AMR is poorly documented in animals in low-income and middle-income countries, and this may be partly due to the absence of systematic surveillance systems and large epidemiological studies[14].

The domestic pig (*Sus scrofa domesticus*) is one of the most important human food-producing animals, as over 40% of all meat consumed on earth comes from pigs[15]. Geographically, pigs are farmed in many regions around the world, but more than 60% of the world's pigs are found in Asia, particularly eastern Asia and southeastern Asia[16]. Recently, the analysis of *E. coli* isolates from pigs and other animals by antimicrobial susceptibility testing (AST) in combination with whole-genome sequencing (WGS) has facilitated a better understanding of AMR development and dissemination[17–20]. For example, a recent study showed that plasmid-borne *mcr-1* is responsible for the rapid dissemination of polymyxin resistance in *Enterobacteriaceae* in animals and humans[4]. Another recent study revealed that IncX4, IncI2, and IncHI2 were the common plasmid types carried by *mcr-1*-positive *E. coli* on pig farms in China after the cessation of colistin use as an animal feed additive, and significant increases in IncI2-associated *mcr-1* and a distinct lineage of *mcr-1*-associated IncHI2 were observed after this agent was banned[21]. Based on WGS data, many studies have also shown that *E. coli* sequence type (ST) 10 and its related STs are important AMR-associated genotypes that predominate in swine, and a higher prevalence of plasmid-borne ARGs is found in these types than in others[7,21–25].

China is the largest pig-rearing country in the world and many provinces include low-income and middle-income regions. China is also the top antimicrobial-producing, and pig-producing, and pork-consuming country worldwide[26,27]. Therefore, understanding the current profile of AMR on pig farms in China is of

great significance for global AMR surveillance. However, pig production in China is very complex, involving different sizes of pig farms and production models in various regions. There remains a lack of systemic data reflecting the condition of AMR in the pork production chain in China. Here, we identified AMR on pig farms in most regions of China by investigating *E. coli*, which is a commonly used biomarker of AMR on pig farms[10]. To the best of our knowledge, this is the first time that pig farms in all regions of mainland China have been included in an epidemiological investigation. This study will contribute to understanding the current profile of AMR in pigs and provide a reference for the formulation of AMR control policies for livestock in China.

## Results

**Antimicrobial-resistant phenotypes.** Altogether, a total of 1871 *E. coli* isolates were recovered from pigs or their breeding environments in 31 Chinese provinces between 1 October 2018 and 30 September 2019 (Fig. 1a and Supplementary Fig. 1). The most represented provinces by isolate count were Henan (*n* = 191) and Hubei (*n* = 250), which are the two largest pig-farming provinces in China. Initially, we attempted to include more farms in each of the provinces, but the sudden outbreak of African swine fever (ASF) in late 2018 and coronavirus disease (COVID-19) in late 2019 made it extremely difficult to obtain more samples from additional farms.

AST results showed that 90.54% (1694/1871) of the 1871 isolates were multidrug-resistant (MDR) strains (Fig. 1b). A large proportion of these isolates were resistant to tetracycline (96.26%, 1801/1871), chloramphenicol (82.04%, 1535/1871), moxifloxacin (81.56%, 1526/1871), and trimethoprim/sulfamethoxazole (80.38%, 1504/1871), while a relatively small proportion of the isolates were resistant to colistin (3.79%, 71/1871), carbapenems (imipenem [2.62%, 49/1871], meropenem [2.30%, 43/1871], ertapenem [2.46%, 46/1871]), and broad-spectrum cephalosporins (ceftriaxone [29.56%, 553/1871] and cefepime [14.00%, 262/1871]) (Fig. 1c and Supplementary Fig. 1 and Supplementary Data 1). Many isolates showed resistance to tigecycline (37.31%, 698/1871), but most of them showed minimum inhibitory concentrations (MICs) ranging from 0.5 to 1 µg/ml (92.98%, 649/1871), and a very small proportion of the isolates showed a high level of resistance (MIC ≥ 4 µg/ml; 0.72%, 5/1871) (Supplementary Fig. 1).

The AST results also revealed that resistance to tetracycline, chloramphenicol, moxifloxacin, and trimethoprim/sulfamethoxazole was a common phenotype of the isolates from the pig farms in all 31 provinces of mainland China (Fig. 1c). Carbapenem-resistant isolates were found in seven provinces, but the majority came from Henan Province; isolates resistant to colistin were found in 12 provinces, and most of these isolates were also recovered from Henan Province (Fig. 1c). Tigecycline-resistant *E. coli* were found in 28 provinces, including Tibet, but highly tigecycline-resistant strains (MIC value ≥4 µg/ml) were only identified in Anhui, Hunan, Guizhou, Hebei, and Hubei (Fig. 1c). *E. coli* strains resistant to broad-spectrum cephalosporins (ceftriaxone and cefepime) were isolated from pig farms in 30 and 24 provinces, respectively (Fig. 1c). Tibet was the only region where no strains from pig farms with the abovementioned resistant phenotypes were detected (Fig. 1c). Notably, MDR isolates were identified on pig farms in 31 Chinese provinces, but a relatively high proportion of the MDR isolates were identified on farms in Sichuan (69.44%) and Tibet (42.86%) relative to those from the other provinces. Strikingly, all isolates from pig farms in Beijing and Ningxia were determined to be MDR strains (Figs. 1d and 2a).

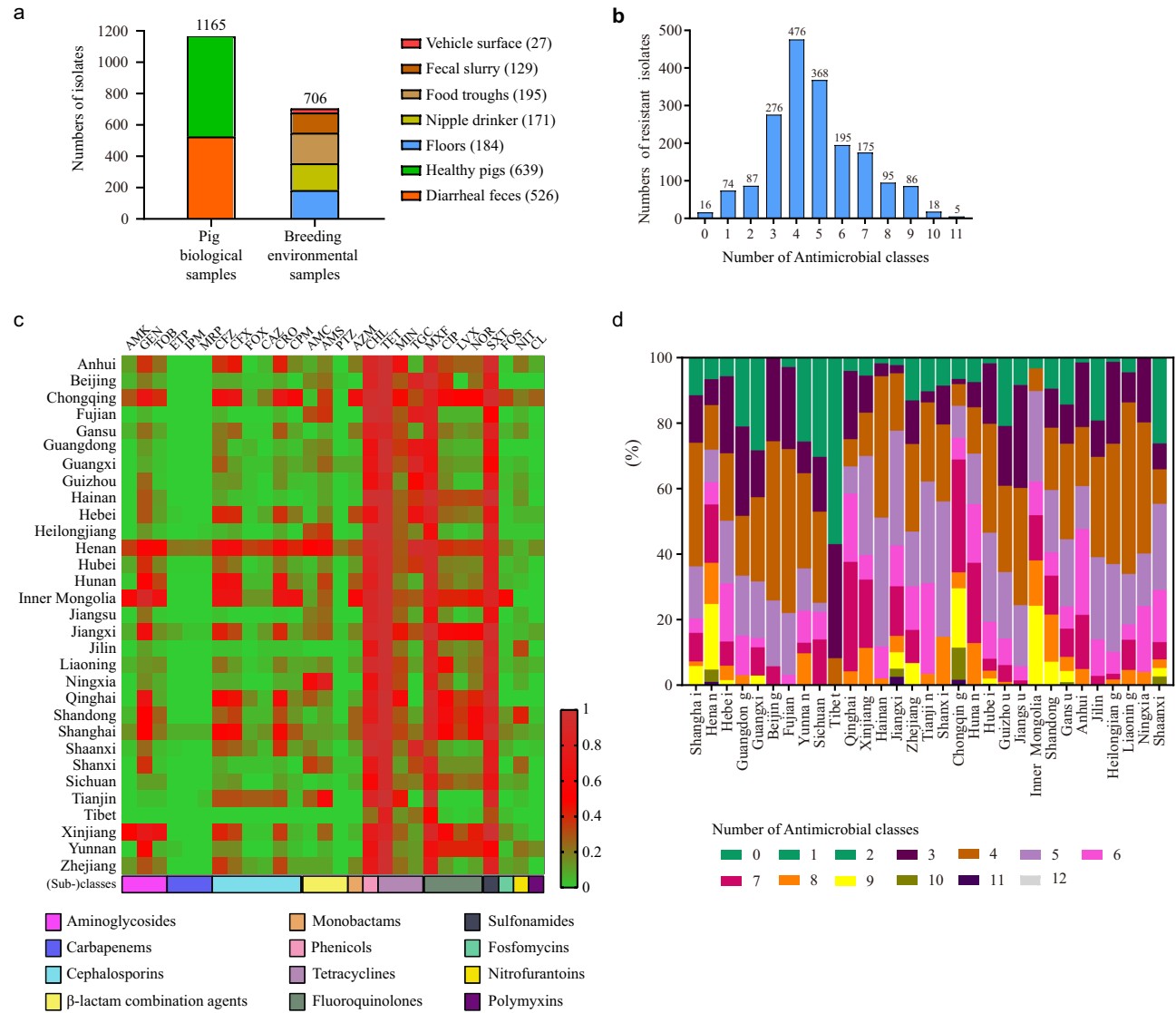

**Fig. 1 Isolation and antimicrobial resistance characteristics of *E. coli* isolates from pig farms in different provinces of China. a** Numbers of *E. coli* isolates recovered from different types of samples collected from Chinese pig farms. Numbers in brackets refer to the numbers of *E. coli* isolates recovered from these samples; **b** Numbers of *E. coli* isolates resistant to different antibiotic classes tested; **c** Heatmap showing the percentage of farm-originating *E. coli* isolates resistant to each of the antibiotics tested; **d** Percentages of farm-originating *E. coli* isolates resistant to different antibiotic classes in different provinces of mainland China. AMK amikacin, GEN gentamicin, TOB tobramycin, IPM imipenem, MRP meropenem, ETP ertapenem, CFZ cefazolin, CFX cefuroxime, FOX cefoxitin, CAZ ceftazidime, CRO ceftriaxone, CPM cefepime, AMC amoxicillin/clavulanate, AMS ampicillin/sulbactam, PTZ piperacillin/tazobactam, AZM aztreonam, CHL chloramphenicol, TET tetracycline, MIN minocycline, TGC tigecycline, MXF moxifloxacin, CIP ciprofloxacin, LVX levofloxacin, NOR norfloxacin, SXT trimethoprim/sulfamethoxazole, FOS fosfomycin, NIT nitrofurantoin, CL colistin.

## O-serogroups, sequence types, and virulence factor-encoding genes

To understand the genomic characteristics of the drug-resistant *E. coli* isolates on Chinese pig farms, we performed Illumina sequencing on 515 MDR isolates recovered in this study (Supplementary Data 2). In silico serotyping using the WGS data identified 101 kinds of O-serogroups, among which O9a ($n = 50$), O101 ($n = 46$), and O8 ($n = 39$) were the predominant types (Supplementary Data 2). Notably, many O-serogroups that might have public health significance were also identified, including O101, O128ac, O11, O136, O28ac, O103, O149, O15, O45, O125ab, O9, O115, O159, O73, O25, O26, O29, O6, O8, O80, O143, O148, O153, O157, O166, O167, O78, O86, and O91. Multilocus sequence typing (MLST) analyses of the 515 isolates demonstrated significant diversity, with the isolates being assigned to 118 distinct STs, except for 41 isolates with for which

an ST could not be confidently identified (Supplementary Data 2). Among the 118 identified STs, ST10 ($n = 52$), ST101 ($n = 39$), and ST48 ($n = 22$) were the most prevalent (Fig. 2b).

We next analyzed the virulence factor-encoding genes (VFGs) carried by the sequenced *E. coli* isolates. According to the prediction results, each of the isolates contained numerous VFGs (VFG numbers ranged from 78 to 284; Supplementary Data 3). Among these genes, *astA*, *eae*, *east1*, *ecpABCDER*, *efa1*, *eltAB*, *escCDFJNRSTUV*, *espABD*, *estIa*, *paa*, *pic*, *stb*, *stx*$_{2eB}$, *tir*, and *toxB* were of particular note (Supplementary Data 3). These genes encode important adherence factors and/or toxins, and *E. coli* isolates possessing these VFGs are assigned as pathogenic *E. coli*[28]. In particular, these VFGs were identified in MDR *E. coli* isolates with the abovementioned O-serotypes that are of public health significance (Supplementary Data 3).

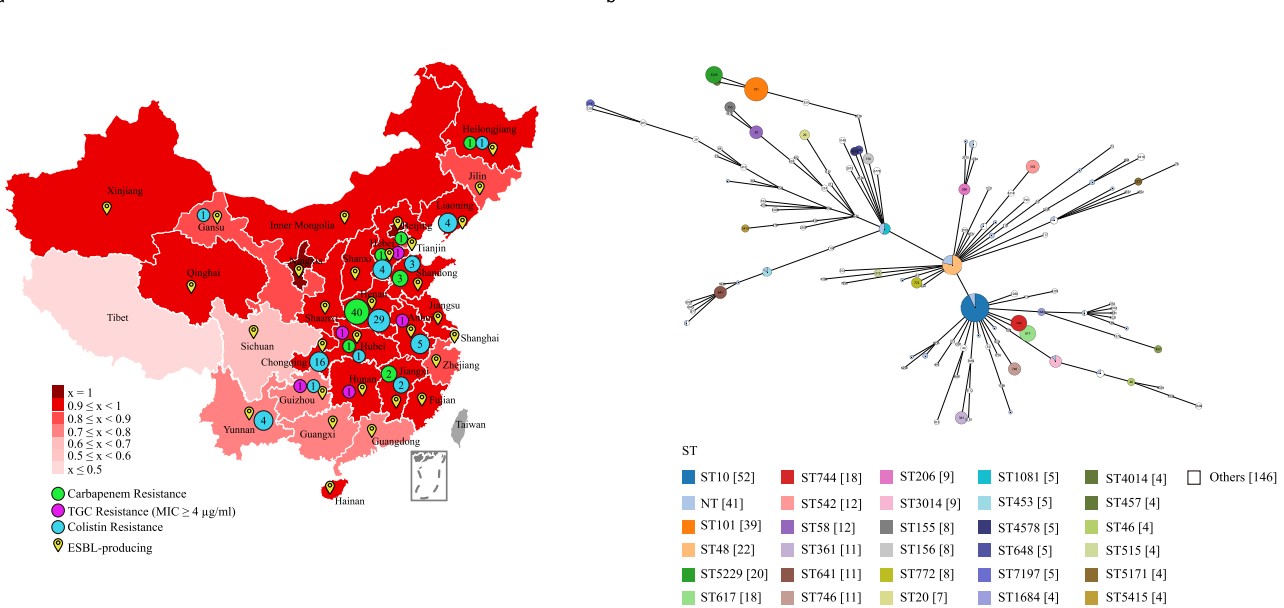

**Fig. 2 Distribution and sequence types of multidrug-resistant *E. coli* isolates from Chinese pig farms. a** A map of China showing the percentages of multidrug-resistant *E. coli* isolates from pig farms in different provinces of China ("x" refers to percent multidrug-resistant isolates); different colors represent percentages; small circles in green, purple and blue show the distribution of carbapenem-resistant, colistin-resistant, and tigecycline-resistant *E. coli* isolates, respectively; numbers in small circles represent the numbers of carbapenem-resistant, colistin-resistant, and tigecycline-resistant *E. coli* isolates, respectively, in different provinces; ESBL-producing *E. coli* isolates are also indicated. **b** Analysis of the minimum spanning tree of the 515 sequenced *E. coli* isolates based on the sequence type. TGC tigecycline, ESBL extended-spectrum beta-lactamase, MLST multilocus sequence typing, ST sequence type.

**Genomic associations with antimicrobial resistance phenotypes.** To understand the genomic basis of the antimicrobial resistance phenotypes of the sequenced *E. coli*, we searched for the presence of ARGs in the genome sequences. This approach led to the detection of 109 ARGs (Supplementary Data 4). Of particular note was the identification of four ESBL genes ($bla_{CTX-M}$, $bla_{TEM}$, $bla_{OXA}$, and $bla_{SHV}$), seven fluoroquinolone-resistance genes (*aac(6')-Ib-cr*, *oqxA*, *oqxB*, *qepA*, *qnrB*, *qnrD*, and *qnrS*), three carbapenem-resistance genes ($bla_{NDM-1}$, $bla_{NDM-5}$, and $bla_{NDM-7}$), and two colistin-resistance genes (*mcr-1* and *mcr-3*). Resistance phenotypes generally corresponded to carriage of one or more genes known to confer that phenotype (Fig. 3). For example, ESBL genes were associated with resistance to broad-spectrum cephalosporins. In addition, only carbapenem-resistant isolates contained the three carbapenem-resistance genes, and only colistin-resistant isolates carried the three *mcr* genes. Although different tetracycline resistance genes (*tetA*, *tetB*, *tetC*, and *tetM*) were found in tetracycline-resistant isolates, *tetX4* was only identified in isolates with a high level of resistance to tigecycline (MIC value ≥4 μg/ml). Notably, over 15% of NDM-producing isolates carried *mcr-1*, and only one isolate contained *mcr-1* and *tetX4* (Fig. 3).

**Class I integrons and plasmids.** To understand the genetic basis of ARG dissemination, we first detected the presence of the class 1 integrase gene *intI1* in the genomes of the isolates, and 491 isolates were found to harbor *intI1* (Supplementary Data 5). More than half of the isolates (61.71%, $n = 303$) possessed a single *intI1* gene, and 52 isolates harbored three *intI1* genes. Interestingly, more than half (71.15%, $n = 37$) of the ST10 isolates harbored only one *intI1* gene, while only three ST10 isolates possessed three *intI1* genes.

We next investigated the presence of plasmids in the whole-genome sequences of these drug-resistant isolates, and a total of 53 groups of plasmid replicons were detected (Supplementary

Fig. 2 and Supplementary Data 6). A ColRNAI-type plasmid was present in most of the isolates ($n = 272$), after which IncFIB ($n = 266$) and IncX1 ($n = 236$) were the most common (Supplementary Fig. 2a). These three types of plasmids were also carried by most of the ESBL genes carrying isolates [ColRNAI (260/495), IncFIB (257/495), IncX1 (228/495)], quinolone-resistant isolates [ColRNAI (248/473), IncFIB (257/473), and IncX1 (228/473)], and tetracycline-resistant isolates [ColRNAI (270/510), IncFIB (265/510), and IncX1 (235/510)]. Contig-mapping of all $bla_{NDM}$-carrying isolates ($n = 45$) identified 34 groups of plasmids (Fig. 3 and Supplementary Data 6). Most of them carried IncFII(pHN7A8) (27/45), ColRNAI (25/45), IncX3 (24/45), Col(MG828) (21/45), and Col156 (20/45). A total of 41 groups of plasmids were identified in the *mcr*-carrying isolates ($n = 69$) (Fig. 3 and Supplementary Data 6), and most of them carried ColRNAI (50/69), IncX1 (39/69), RepA (36/69), IncHI2 (36/69), and IncHI2A (36/69).

We then generated the complete genome sequences of the XD33 isolate producing both NDM and MCR (GenBank accession no. JAENDM000000000) through Oxford Nanopore Sequencing (ONT). This approach identified an 85.9 kb IncFII plasmid (designated pXD33-05) carrying the $bla_{NDM-1}$ gene and a 33.3 kb IncX4 plasmid (designated pXD33-06) carrying the *mcr-1* gene (Fig. 4a, b). Strikingly, other isolates that co-produced NDM and MCR ($n = 6$) displayed high sequence homology across the vast majority of these plasmid backbones, strongly suggesting carriage of highly similar plasmids (Supplementary Fig. 3). BLAST analysis of the pXD33-05 sequence showed that the backbone of pXD33-05 (plasmid excluding the MDR element) displayed 99% homology to that of pHNEC55 (GenBank accession no. KT879914), a plasmid carried by *E. coli* HNEC55 (Fig. 4d and Supplementary Fig. 2b), a carbapenem-resistant strain also obtained from pig farms in Henan Province[29]. Interestingly, plasmids with similar backbones carrying different MDR cassettes have also been reported previously, including

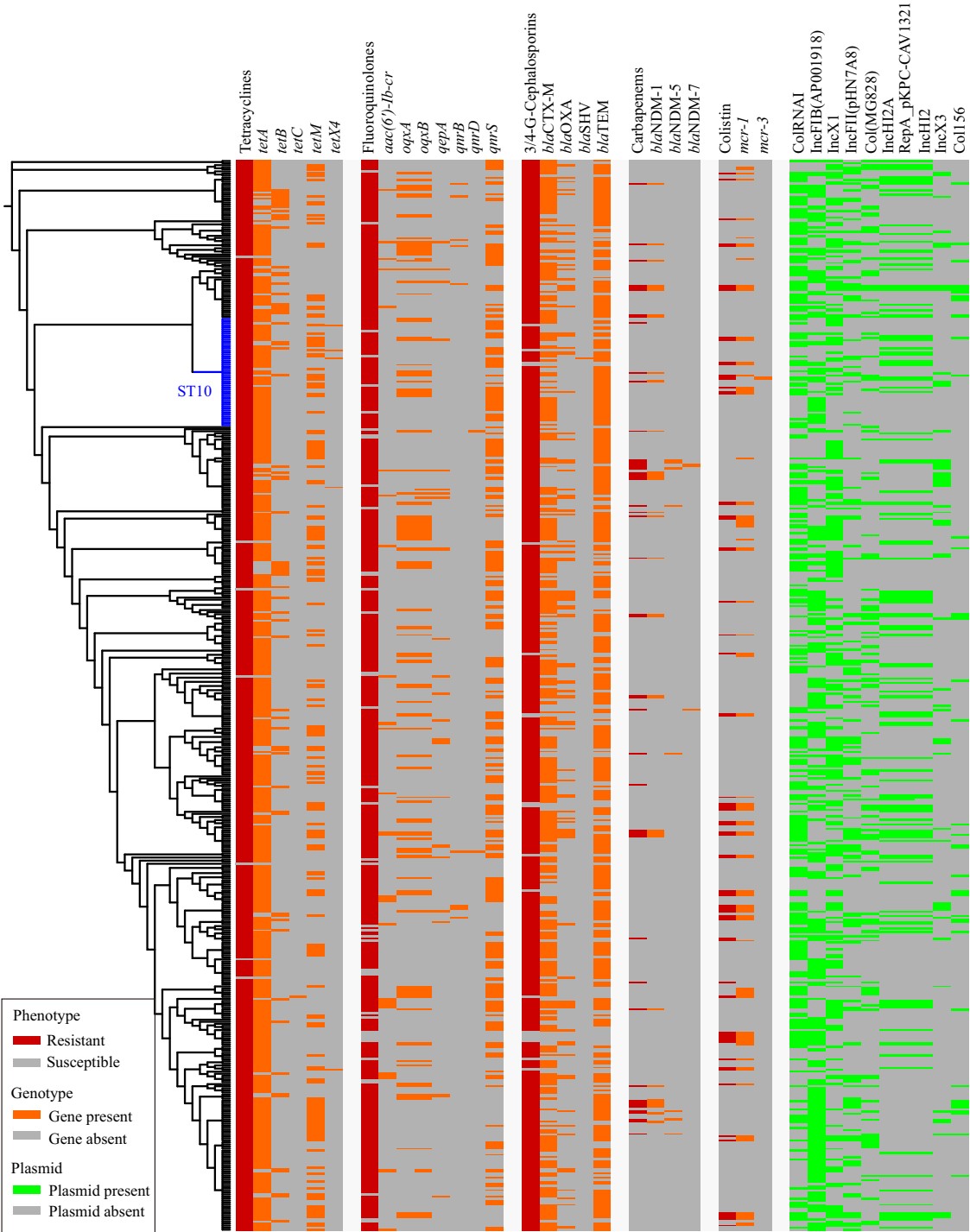

**Fig. 3 Distribution of antimicrobial resistance phenotypes, antimicrobial resistance genes and plasmid replicons.** The phylogenetic tree was generated based on the concatenated multilocus sequence typing (MLST) alleles using fastMLST v0.0.15, followed by multiple sequence alignment using MAFFT v7.407 and phylogenetic inference using FastTree. The tree was visualized using the Interactive Tree Of Life (iTOL v.5). Raw data for the MLST-based tree and its annotation codes are provided in Supplementary Data 11.

plasmid pHNHNC02 (GenBank accession no. MG197497), which was recovered from an *E. coli* strain from chickens, and plasmid pKPC-L388 (GenBank accession no. CP029225), which was carried by a clinical *K. pneumoniae* isolate from humans (Fig. 4d). The MDR elements of pXD33-05 consisted of two ARG cassettes, including a 7.6 kb cassette harboring a bleomycin resistance gene, the aminoglycoside resistance gene *aph(3′)-VI* and *bla*$_{NDM-1}$ and a 2.4 kb cassette harboring the aminoglycoside resistance gene

*rmtB* and the ESBL-encoding gene *bla*$_{TEM-1B}$. BLAST analysis of the pXD33-06 sequence showed that pXD33-06 displayed 99% homology to the plasmid pWI2-mcr (GenBank accession no. LT838201) carried by an *E. coli* clinical isolate from humans, plasmid pSH15G2169 (GenBank accession no. MH522417) from a clinical *Salmonella enterica* isolate from humans, and plasmid 16BU137_mcr-1.1 (GenBank accession no. MT316509) carried by a clinical *K. pneumoniae* isolate from humans. The MDR

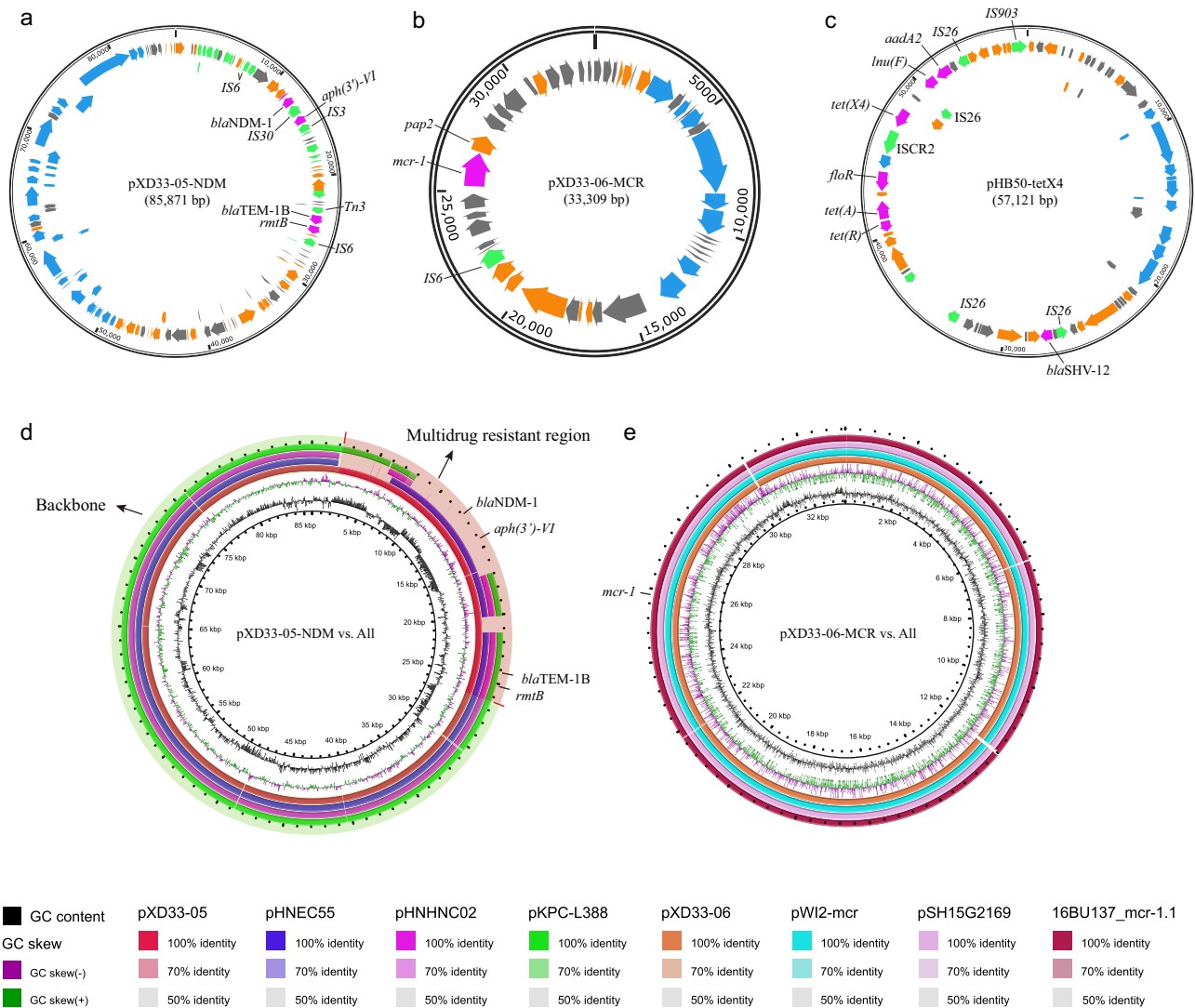

**Fig. 4 Genetic characteristics of the $bla_{NDM}$-carrying, $mcr$-carrying, and/or $tetX4$-carrying plasmids identified in this study. a** Circle map showing the structure of the $bla_{NDM}$-carrying plasmid pXD33-05; **b** Circle map showing the structure of the $mcr$-carrying plasmid pXD33-06; **c** Circle map showing the structure of the $tetX4$-carrying plasmid pHB50-tetX4; arrows in purple, green, gray, orange, and blue refer to antimicrobial resistance genes, mobilization-associated genes (including insertion elements), hypothetical protein-encoding genes, functional protein-encoding genes, and type IV pilus formation-related genes, respectively; **d** Circle map showing sequence alignments of pXD33-05 and those plasmids with a backbone shared with other *Enterobacteriaceae* bacteria; circles from inside to outside indicate the GC content of pXD33-05 (*circle 1*), the GC skew of pXD33-05 (*circle 2*), pXD33-05 (*circle 3*), the *E. coli* plasmid pHNEC55 (GenBank accession no. KT879914; *circle 4*), the *E. coli* plasmid pHNHNC02 (GenBank accession no. MG197497; *circle 5*), and the *K. pneumoniae* plasmid pKPC-L388 (GenBank accession no. CP029225; *circle 6*); the plasmid backbone regions are highlighted in light green, while the multidrug resistance regions are highlighted in light red; nucleotide identities are shown with different colors below the maps; **e** Circle map showing the sequence alignments of pXD33-06 and those plasmids with a backbone shared with other *Enterobacteriaceae* bacteria; circles from inside to outside indicate the GC content of pXD33-06 (*circle 1*), the GC skew of pXD33-06 (*circle 2*), pXD33-06 (*circle 3*), the *E. coli* plasmid pWI2-mcr (GenBank accession no. LT838201; *circle 4*), the *Salmonella* plasmid pSH15G2169 (GenBank accession no. MH522417; *circle 5*), and the *K. pneumoniae* plasmid 16BU137_mcr-1.1 (GenBank accession no. MT316509; *circle 6*); nucleotide identities are shown with different colors below the map.

elements of pXD33-06 harbored only the *mcr-1* gene (Fig. 4e and Supplementary Fig. 2c).

ONT sequencing of two highly tigecycline-resistant isolates, HB50 and SY36, revealed that *tetX4* was carried by an IncX1 plasmid in both isolates (Fig. 4c). BLAST analyses revealed that the sequence of this IncX1 plasmid displayed 99% homology to plasmid pYY76-1-2 (GenBank accession no. CP040929), which is also a *tetX4*-bearing plasmid carried by an *E. coli* isolate from the feces of a cow[30]. Plasmids with similar backbones were also reported previously, including the pEC931_tetX plasmid (Gen-Bank accession no. CP049121), which was recovered from a clinical *E. coli* isolate from humans. In all identified *tetX4*-

carrying elements, *tetX4* was adjacent to an IS*CR2* element (Supplementary Fig. 2d). Plasmid conjugation experiments revealed that most of the $bla_{NDM}$-carrying, $mcr$-carrying, and *tetX4*-carrying plasmids were conjugative and conferred phenotypes of carbapenem, colistin, and high tigecycline (MIC value ≥4 µg/ml) resistance in their bacterial recipients (Supplementary Data 7).

**Relationships with human commensal *E. coli*.** To determine the genetic propensity of the drug-resistant isolates to spread into the human sector, the genetic relatedness of the 515 MDR *E. coli*

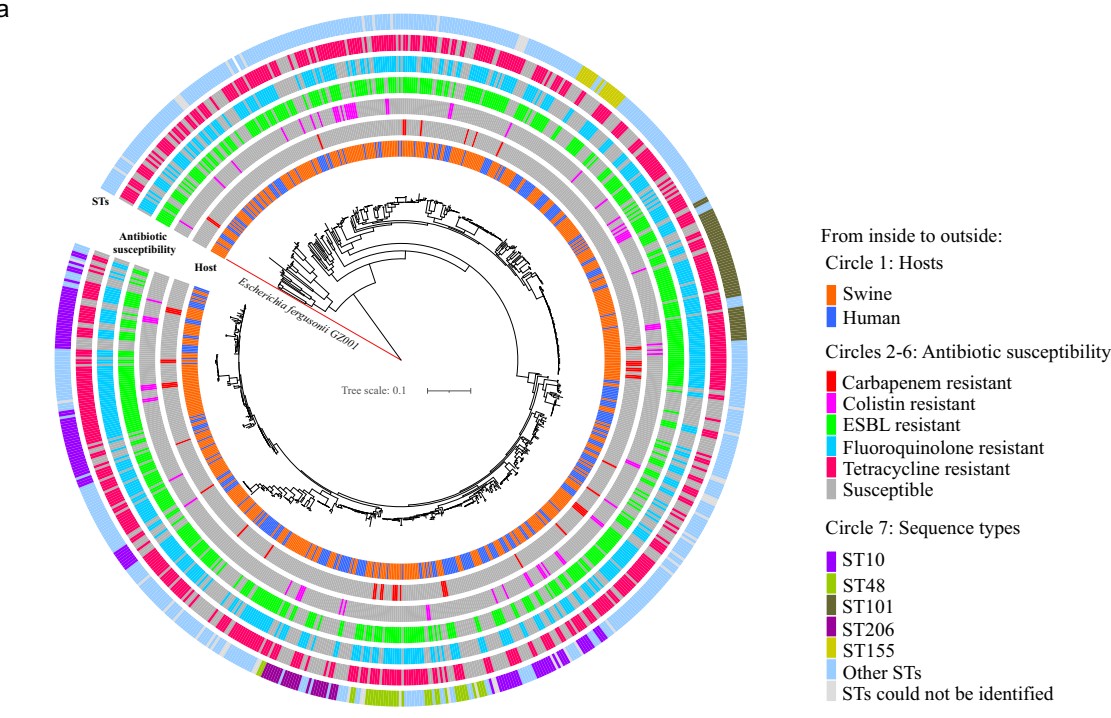

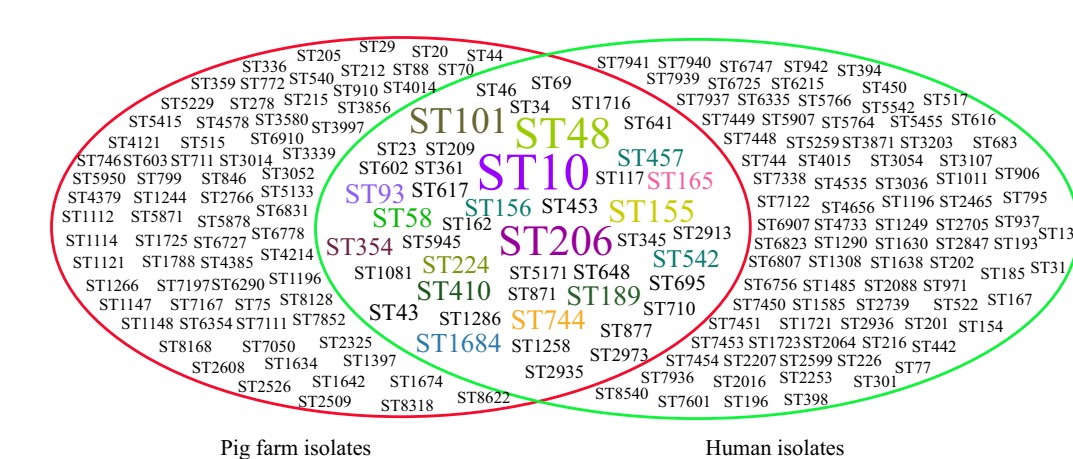

**Fig. 5 Phylogenetic trees of 515 multidrug-resistant *E. coli* isolates from pig farms together with 287 publicly available draft genomes of human commensal *E. coli* across China. a** Circular tree showing the phylogenetic relationship of the 802 *E. coli* isolates. The tree was generated based on the core genome SNPs using the snippy-multi program implemented in Snippy v 4.4.0 (https://github.com/tseemann/snippy), and was rooted on the *E. fergusonii* sequence. The snippy based tree was inferred using FastTree2, and was visualized using the Interactive Tree Of Life (iTOL v.5). Raw data for the Snippy tree and its annotation codes are provided in Supplementary Data 12; **b** Venn diagram showing the shared and unique sequence types between pig farm-originating *E. coli* and human commensal *E. coli*.

isolates from pig farms in China to 287 publicly available draft genomes of human commensal *E. coli* (Bioproject no. PRJNA400107) was analyzed[31]. The phylogenetic analysis of the 802 *E. coli* isolates showed that the 515 MDR *E. coli* isolates from pig farms were closely related to the 287 human *E. coli* strains (Fig. 5a). Single nucleotide polymorphism (SNP) analyses revealed that a large proportion of the pig farm isolates showed high genetic similarity (228/515, as few as three SNPs) to the

human isolates (Fig. 5 and Supplementary Data 8). MLST of the 287 human commensal isolates also revealed significant diversity of STs, where the isolates were assigned to 132 distinct STs, with the exception of 11 isolates with novel STs (Fig. 5b and Supplementary Data 9). Interestingly, ST10 was the most common ST and was carried by 44 human isolates (15.33%). Among the 132 distinct STs, 45 STs were also shared by the 515 drug-resistant isolates from pig farms (Fig. 5b).

## Discussion

In this study, we present a nationwide surveillance report encompassing isolates from pig farms in all of China's mainland provinces. While a large number of similar studies have been conducted on different Chinese pig farms[32–34], these studies have been limited at a spatial level. Most of them were conducted in regions (e.g., Henan, Sichuan, Hubei, Hunan, and Shandong) with large pig populations, where the Chinese pig-farming industry has been developed. In regions, where small pig populations are reared (e.g., Qinghai and Tibet), small amounts of associated data have been reported[14]. To the best of our knowledge, this is the first study to investigate the AMR phenotypes of *E. coli* isolates from pig farms in all provinces of mainland China during the same period.

Our AST results suggest a worrisome AMR situation on pig farms, as evidenced by the common recovery of MDR *E. coli* isolates from both pigs and their breeding environments on farms in different provinces, including Tibet, Xinjiang, and Qinghai. This worrisome situation is widely accepted to be the result of antibiotic overuse and misuse in the Chinese pig industry[26,35]. During recent decades, with the rapid increase in the economy, the production of meat, eggs, and milk has rapidly increased in China; this is especially true for pork, the main source of animal protein for most Chinese people[26,36]. To meet the increasing demand for pork, both the number and the size of pig farms have increased markedly, and massive amounts of antibiotics are used in the country to support the rapid increase in pig production[26].

Our AST results also demonstrated that *E. coli* isolates from Chinese pig farms were commonly resistant to several specific antimicrobial classes, including sulfonamides, tetracyclines, fluoroquinolones, macrolides, and β-lactams (excluding carbapenems). In addition, ARGs associated with resistance to these antimicrobial classes were also commonly identified in isolates from the farms. This may be because all of these antimicrobial classes have been extensively used on Chinese pig farms, particularly fluoroquinolones and β-lactams, which account for more than half of the application of these agents[27,37,38]. A previous study reported that total tetracycline concentrations in manure and soil samples from several Chinese pig farms were as high as 15.2 and 0.78 mg kg$^{-1}$, while the highest mean concentrations of ofloxacin and enrofloxacin observed in samples collected from these farms were as high as 335 µg kg$^{-1}$ and 96.0 µg kg$^{-1}$, respectively[39]. In the present study, although farmers on the majority of farms included in our investigation were not willing to disclose the antibiotics used in their facilities, several disclosed that antibiotics belonging to the sulfonamide (e.g., sulfamonomethoxine), tetracycline (e.g., chlortetracycline, doxycycline, and oxycycline), fluoroquinolone (e.g., enrofloxacin), macrolide (e.g., ivermectin and kitasamycin tartrate), and β-lactam (e.g., cefotaxime, penicillin, and amoxycillin) classes were used on their farms for growth promotion or disease therapy.

Although the use of chloramphenicol in animal-producing facilities was banned in China in 2002[40], chloramphenicol-resistant isolates were still frequently recovered from pig farms in different provinces in this study. This might be because florfenicol, a member of the same class as chloramphenicol, is still in use[26] and may be responsible for the maintenance of chloramphenicol resistance in *E. coli* from Chinese pig farms. Another noteworthy finding was that colistin-resistant isolates were recovered from pig farms in 12 provinces in low proportions (3.79%). This relatively low proportion of recovery is attributed to China's policy banning the use of colistin in agriculture in 2017, which has had a significant effect on reducing colistin resistance in both animals and humans[21,41]. Although many *E. coli* isolates from pig farms in China showed resistance to tigecycline (37.31%, $n = 698$), most of them displayed a low level of resistance (MIC

values ranging from 0.5 to 1 µg/ml; 92.98%, $n = 649$). This may be because the breakpoint used for the interpretation of tigecycline resistance recommended by the European Committee on Antimicrobial Susceptibility Testing (EUCAST) is low (0.5 µg/ml, EUCAST document v 8.1). A phenotype of low tigecycline resistance may be conferred by tetracycline resistance genes (e.g., *tetA*, *tetB*, *tetC*, and/or *tetM*[42]) since these genes are commonly carried by *E. coli* from animals.

Our investigation also showed the worrisome situation of the presence carbapenem-resistant isolates on pig farms in seven provinces, all of which were NDM-producing isolates. The recovery of these isolates was very unusual and unexpected because carbapenem antibiotics are not approved for use in livestock in China according to the official policy. A possible explanation is that some pig farms might use these antibiotics as a therapy without receiving approval because infections caused by MDR bacteria are very common on livestock farms[2,32,33,43], and carbapenem antibiotics are considered last-resort effective options for treating MDR bacterial infections[44]. However, we did not obtain the required information since the farmers on these farms were not willing to disclose the antibiotics used at their facilities. Another possible reason might be contamination in the in-house environment, as a recent study showed a high detection rate (26.8–31.4%) of $bla_{NDM}$ in environmental samples, except in air, after standard cleaning and disinfection during the vacancy period on a Chinese poultry farm[43]. Interestingly, the reported rates of carbapenem-resistant *E. coli* in human health care facilities in provinces from which farm-originating carbapenem-resistant isolates have been recovered are higher than those in provinces, where farm-originating carbapenem-resistant isolates have not been recovered, according to the China Antimicrobial Resistance Surveillance System (CARSS)[45,46]. Therefore, it can be hypothesized that carbapenem-resistant bacteria or $bla_{NDM}$ from human health care facilities may lead to contamination by these hazardous agents in environments (e.g., water, air, etc.); this contamination may in turn spread to livestock farms, resulting in the emergence of carbapenem-resistant bacteria in animals. In addition, a third possibility worth considering is that the plasmid and/or isolate was co-selected by the use of another antimicrobial that may be present in the plasmid and/or isolate. However, further investigations are necessary to verify the above three assumptions.

Our WGS analysis identified several O-serogroups that are frequently associated with *E. coli* pathotypes[47,48]. Importantly, several marker VFGs of *E. coli* pathotypes (e.g., *astA*, *eae*, *east1*, *stb*, $stx_{2eB}$, and *toxB*[47,48]) were also identified in *E. coli* isolates belonging to these O-serogroups. However, only 36.64% (66/181) of these isolates were recovered from the feces of diarrheal pigs, and the remaining isolates came from environmental samples. The widespread presence of these MDR *E. coli* on pig farms represents a great health risk and should receive more attention. Our WGS analysis also demonstrated that ST10 ($n = 52$), ST101 ($n = 39$), and ST48 ($n = 22$) were widely distributed among the MDR *E. coli* isolates from pig farms. It should be noted that all three of these STs, particularly ST10, have been reported to be important AMR-associated genotypes predominating in pigs[7,21–25]. Of particular note, the ST10 clone has been revealed to be a reservoir of ARGs and mobile genetic elements, such as class 1 integrons and plasmids[24,25]. It should also be noted that these three STs have been frequently identified in *E. coli* pathotypes (ETEC/EPEC/STEC) from pigs with diarrhea[49]. Accordingly, these three STs were commonly identified among the 207 isolates from the feces of diarrheal pigs [ST10 ($n = 28$), ST101 ($n = 14$), and ST48 ($n = 8$)] in this study. Notably, these three STs were also predominantly identified STs among the 287 human commensal isolates analyzed in this study. Furthermore, these

findings are in agreement with those of a recent study suggesting that *E. coli* clonal complex 10, including ST10, may be associated with fecal carriage in humans[24].

It has been documented that *E. coli* has a great capacity to accumulate ARGs, mostly through horizontal gene transfer[7]. Accordingly, a large number of acquired ARGs were detected in the genomes of the isolates recovered in this study. Along with the detection of multiple ARGs, a large number of the sequenced isolates were found to carry the class 1 integrase gene *intI1*. This gene is always linked to genes conferring resistance to antibiotics, disinfectants, and heavy metals, and it could be used as a proxy for anthropogenic pollution[50]. In addition to the class 1 integrase, we detected many putative plasmids in *E. coli* isolates from pig farms in China. Several groups of these plasmids (e.g., IncA/C, IncB/O, IncFIB, IncX1, IncFIIA, IncX2, and IncY) are capable of carrying sequences conferring transfer, MDR, and virulence functions in a broad host range[51]. The presence of those plasmids may accelerate the dissemination of ARGs. We also identified several groups of plasmids that were rarely associated with the spread of specific ARGs. For example, the IncX3 plasmid has been reported to account for the majority of $bla_{NDM}$ carriage in livestock farms[43,52]. However, we also identified an IncFII-type-$bla_{NDM}$-carrying plasmid, and this type of plasmid was observed in all *E. coli* isolates coproducing NDM and MCR. In addition, the *mcr*-carrying plasmid identified in *E. coli* isolates coproducing NDM and MCR in this study was different from the plasmid mediating the dissemination of *mcr* in China reported recently[4]. The presence of these plasmids makes the dissemination of $bla_{NDM-1}$ and/or *mcr* more heterogeneous. It is worth noting that the elements mediating the spread of the high tigecycline resistance gene *tetX4* in *E. coli* isolates from pig farms in this study were highly homologous to those reported previously[3,31,53], suggesting that the dissemination of *tetX4* might not be as heterogeneous as that of $bla_{NDM}$ and *mcr*. However, continuous monitoring should be undertaken in the future. Most notably, the MDR *E. coli* isolates from pig farms displayed very close relatedness to *E. coli* strains from humans in China, and most of them shared the same STs as the human isolates. Many of the *E. coli* isolates obtained from pig farms in this study differed by fewer than 100 SNPs from the human-originating *E. coli* strains. These findings suggest a very high genetic propensity of farm-sourced MDR *E. coli* to spread into humans.

Although this work is limited by our inability to include samples from more pig farms in different provinces of China due to the outbreaks of ASF and COVID-19, our sample collection still covers pig farms in all provinces of China. We characterized the resistance phenotypes and population genomics of *E. coli* on pig farms in China on a national scale for the first time. Our results revealed a worrisome situation of AMR on pig farms in China, and there is still a long way to go to reduce AMR in livestock in China. The Chinese government has implemented a series of actions to address the worrisome AMR conditions in animal husbandry. Notably, the Ministry of Agriculture and Rural Affairs (MARA) issued a policy banning the addition of antibiotics to feed to promote animal growth on July 1, 2020 (MARA Announcement No. 194, 07-10-2019). In this study, we also systematically revealed the distribution of O-serogroups, sequence types, ARGs, VFGs, and putative plasmids of MDR *E. coli* on pig farms in different provinces of China. These data will provide comprehensive insights to help understand AMR on pig farms, and may also be beneficial for the development of governmental policies to reduce AMR in the Chinese pig industry. Notably, we also identified many MDR *E. coli* with potential pathogenicity to humans, and most importantly, we found that pig farm-originating MDR *E. coli* showed a very high genetic propensity to spread humans. The persistence and dissemination

of these isolates present important health risks and should receive more attention.

## Methods

**Sample collection, identification, and antimicrobial susceptibility testing**. Between 1 October 2018 and 30 September 2019, an AMR surveillance project was established based on *E. coli* isolated from pig farms in China. In each of the 31 provinces of mainland China, 2–3 different pig farms were randomly selected to collect swabs of fresh feces and rectal swabs of pigs (with approximately 40 samples per farm) as well as swabs of drinking and fecal slurries, floors, and troughs (at each sampling point, at least three samples per farm were collected). All samples were immediately shipped with dry ice for *E. coli* isolation. The need for ethical approval has been waived by the Animal Management and Ethics Committee of Huazhong Agricultural University because the sample collection does not involve in the use of animal tissues. The reference ID number for the waiver is HZAUSW-2018-023. The swabs were incubated in Luria Bertani (LB) broth (Sigma-Aldrich, MO, USA) at 37 °C and 180 rpm overnight. Thereafter, the swab cultures were streaked on MacConkey agar and incubated at 37 °C for 16 h. Single colonies with similar morphological characteristics (small red/pink round colonies) were picked and confirmed by the PCR amplification of the 16S rRNA gene with primers (F: 5′-GAAGCTTGCTTCTTTGCT-3′, R: 5′-GAGCCCGGGGATTTCACAT-3′) reported previously[54]. PCR assays amplifying seven housekeeping genes of *E. coli* (*adk*, *fumC*, *gyrB*, *icd*, *mdh*, *purA*, and *recA*)[55] were set up for double confirmation.

The antimicrobial susceptibility of *E. coli* isolates was evaluated by determining the MIC values of different kinds of antibiotics in the bacterium following the recommended microbroth dilution protocol (CLSI M100, 28th Edition) of the Clinical & Laboratory Standards Institute (CLSI, United States). A total of 28 types of antibiotics (purchased from MedChemExpress [MCE], Monmouth Junction, NJ) belonging to the aminoglycosides (amikacin, gentamicin, and tobramycin), carbapenems (imipenem, meropenem, and ertapenem), cephalosporins (cefazolin, cefuroxime, cefoxitin, ceftazidime, ceftriaxone, and cefepime), β-lactam combination agents (amoxicillin/clavulanate, ampicillin/sulbactam, and piperacillin/tazobactam), monobactams (aztreonam), phenicols (chloramphenicol), tetracyclines (tetracycline, minocycline, and tigecycline), fluoroquinolones (moxifloxacin, ciprofloxacin, levofloxacin, norfloxacin), sulfonamides (trimethoprim/sulfamethoxazole), fosfomycins (fosfomycin), nitrofurantoins (nitrofurantoin), and polymyxins (colistin) were included in the tests. The results were interpreted using the CLSI breakpoints (CLSI M100, 28th Edition). If the CLSI breakpoint was not available, the European Committee on Antimicrobial Susceptibility Testing (EUCAST) breakpoint (v 8.1) was used for interpretation. Each antibiotic was tested with three duplicates. *E. coli* ATCC 25922 was used for quality control.

**Whole-genome sequencing and data availability**. All *E. coli* strains with phenotypes of resistance to one of the tested carbapenems, broad-spectrum cephalosporins (ceftriaxone and cefepime), tigecycline, or colistin were selected for NGS. Bacterial genomic DNA was extracted by using a commercial DNA Kit (TIANGEN, Beijing, China). The quality and concentration of the bacterial genomic DNA were evaluated via electrophoresis on a 1% agarose gel and analysis on a NanoDrop2000 system (Thermo Scientific, Waltham, MA, USA) and a Qubit 4 Fluorometer (Thermo Scientific, Waltham, USA). Libraries were constructed based on the qualified DNA by using a NEBNext Ultra™ II DNA Library Prep Kit (New England BioLabs, Ipswich, USA) and sequenced on the NovaSeq 6000 platform using the paired-end 150 bp sequencing protocol (Novogene, Beijing, China). Raw reads with low quality were removed as described previously[56]. High-quality reads were de novo assembled with SPAdesv3.9.0 to generate genome contigs.

The complete genome sequences of $bla_{NDM-1}$-carrying, *mcr-1*-carrying, and/or *tetX4*-carrying plasmids were generated by ONT sequencing in combination with Illumina technology. Plasmid DNA was extracted using the phenol-chloroform protocol combined with Phase Lock Gel tubes (Qiagen GmbH), analyzed by agarose gel electrophoresis and quantified on a Qubit® 2.0 system (Thermo Scientific, Waltham, USA). Libraries for ONT and Illumina sequencing were prepared using an SQK-LSK109 kit and a NEBNext® Ultra™ DNA Library Prep kit, respectively. The prepared DNA libraries were sequenced using the Nanopore PromethION platform and an Illumina NovaSeq PE150 system at Novogene Co. LTD (Tianjin, China). ONT and Illumina short reads were finally assembled and combined using Unicycler v0.4.4 software (https://github.com/rrwick/Unicycler/releases) with the default parameters.

**Bioinformatic analysis**. ResFinder 4.1 (https://cge.cbs.dtu.dk/services/ResFinder/)[57] was used to determine putative acquired antimicrobial resistance genes (ARGs) and putative phenotypes of AMR. PlasmidFinder 2.1 (https://cge.cbs.dtu.dk/services/PlasmidFinder/)[58] was used to determine putative plasmids carried by the sequenced strains. In silico serogroups were determined by SerotypeFinder 2.0 (https://cge.cbs.dtu.dk/services/SerotypeFinder/)[59]. Sequence types were identified with the program mlst in GitHub (https://github.com/tseemann/mlst) incorporating components of the PubMLST database (https://pubmlst.org/)[60]. A minimal spanning tree was constructed using GrapeTree (https://www.grapetree.com/) version 1.5.0[61], an interactive tree visualization program in EnteroBase (https://enterobase.warwick.ac.uk/). A

phylogenetic tree was also reconstructed based on the concatenated MLST alleles using fastMLST v0.0.15[62] (https://github.com/EnzoAndree/FastMLST), followed by multiple sequence alignment using MAFFT v7.407[63] (https://mafft.cbrc.jp/alignment/software/) and phylogenetic inference using FastTree[64] (http://www.microbesonline.org/fasttree/). A phylogenetic tree including 515 porcine isolates, 287 human isolates, and an *Escherichia fergusonii* strain (GZ001, GenBank accession no. JAJOYZ000000000, https://www.ncbi.nlm.nih.gov/nuccore/JAJOYZ000000000) was also generated based on the core genome SNPs using the snippy-multi program implemented in Snippy v 4.4.0[65] (https://github.com/tseemann/snippy), and the tree was rooted on the *E. fergusonii* sequence. The snippy based tree was inferred using FastTree2[64]. The genome sequence of *E. coli* XD35 (ST746, GenBank accession no. CP089142, https://www.ncbi.nlm.nih.gov/nuccore/CP089142) was used as a reference. The size of the full alignment was 5,434,282 bp in length, and the number of core SNPs was 170,888 (Supplementary Data 10). The MLST-based phylogenetic tree and the Snippy tree were visualized using the Interactive Tree Of Life (iTOL v.5, https://itol.embl.de/)[66]. Raw data for the MLST-based tree and its annotation codes are provided in supplementary Data 11, and those for the Snippy tree are provided in supplementary Data 12. The RAST Sever was used for sequence annotation[67]. The average nucleotide identities between two genome sequences were calculated by using the ANI calculator (http://enve-omics.ce.gatech.edu/ani/)[68]. A comparative genome analysis was performed and visualized using the BRIG package (http://brig.sourceforge.net/)[69] and/or the EasyFig package (http://brig.sourceforge.net/)[70]. The draft genomes of 287 human commensal *E. coli* (PRJNA400107) were downloaded from NCBI and included in the phylogenetic analysis in this study[31].

**Plasmid conjugation experiments**. Plasmid conjugation experiments between carbapenem-resistant *E. coli*, colistin-resistant *E. coli*, and/or tigecycline-resistant *E. coli* (donors) and rifampin-resistant *E. coli* C600 (recipient) were performed as described previously[23]. Briefly, bacterial donor and recipient strains at mid-log phase ($OD_{600} = 0.5$–0.6) were mixed at a ratio of 1:3 ($v/v$). The bacterial mixture was spotted on nitrocellulose membranes that were preplated on LB agar. Each of the plates were incubated at 37 °C for 12 h, and the bacteria on the membrane were washed using LB broth followed by shaking at 37 °C for 4 h. Transconjugants were selected on LB agar plates with rifampin (1000 mg/l) plus imipenem (20 mg/l) [to screen carbapenem-resistant transconjugants], rifampin (1000 mg/l) plus colistin (2 mg/l) [to screen colistin-resistant transconjugants], or rifampin (1000 mg/l) plus tigecycline (4 mg/l) [to screen tigecycline-resistant transconjugants]. The antimicrobial susceptibility of the transconjugants was identified using the broth microdilution method as mentioned above.

**Reporting summary**. Further information on research design is available in the Nature Research Reporting Summary linked to this article.

## Data availability

The whole-genome sequences (WGSs) of *E. coli* isolates generated in this study have been deposited in GenBank under accession code bioproject PRJNA688628. GenBank accession numbers are given in supplementary Data 2.

## Code availability

The tree files and codes for tree annotations using iTOL (https://itol.embl.de/) are given in Supplementary Data 11 (for the MLST-based tree in Fig. 3) and Supplementary Data 12 (for the Snippy tree in Fig. 5).

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

## Acknowledgements

The authors sincerely acknowledge our colleagues for sample collection. Our work was supported by the National Key Research and Development Program of China (grant number: 2017YFC1600100), the National Natural Science Foundation of China (grant numbers: 32122086, 31902241), the Natural Science Foundation of Hubei Province (grant number: 2020CFB525), the China Agriculture Research System of MOF and MARA, the China Postdoctoral Science Foundation (grant number: 2018M640719), the Walmart Foundation (Project number: 61626817) and the Walmart Food Safety Collaboration Center. The funders had no role in the study design, data collection, data analysis, data interpretation, or manuscript writing.

## Author contributions

Z.P., H.C., and X.W. conceived and designed the study. Z.P., Z.H., Z.L., X.Z., C.J., and T.L. participated in sample collection and laboratory tests. Z.P. and Z.X. made bioinformatical analyses and data curation. Z.P. and X.W. drafted the first version of the manuscript. Z.P., M.D., C.T., Z.X., B.W., H.C., and X.W. contributed to the discussion and revision of the manuscript. All authors read and approved the submitted version.

## Competing interests

The authors declare no competing interests.
