## [Peer Review File · Nature Communications]

Antimicrobial resistance and population genomics of multidrug-resistant *Escherichia coli* in pig farms in mainland ChinaEditorial Note: Parts of this Peer Review File have been redacted as indicated to remove third-party material where no permission to publish could be obtained.

REVIEWER COMMENTS

Reviewer #1 (Remarks to the Author):

General comments

The study by Peng et al presents an overview of the trends in antimicrobial resistance in pigs China. It is a reasonably large survey with each province represented by approximately 50 isolates. The issue of AMR in animal production in China is important and deserves scrutiny. This study contributes to that objective, on an unprecedented geographical scale by combining phenotypic, and genetic assays. However, other studies have already shed light on this issue.

The study by Peng et al is potentially interesting, albeit challenges concerning the writing quality of the manuscript should be addressed before a more complete assessment of the work done can be conducted (see below). However, a key missing aspect of this study is a comparison with trends in AMR in animals reported by the National Surveillance program of the Ministry of Agriculture. This would considerably strengthen this study, and highlight potential divergences with this surveillance systems. I would also encourage the authors to support the statement made in the results and discussion with statistical test and quantitative evidence.

The abstract is 380 words long instead of (maximum) 150.

The abstract contains a lot of elements of method where the reader would expect to see original results. This manuscript contains (too) many results, and (too) many figure. The authors should reconsider the amount and nature of information to be presented in the main manuscript to facilitate the work of the potential readers.

The written English of the manuscript is of poor quality, and makes the work difficult to follow. Example line 331 : "To the best of our knowledge, this is the first time the AMR phenotypes of E. coli isolates from pig farms in all provinces of mainland China, including Tibet and Qinghai where there were little related data before being reported."

Results:

Could the authors specify what they mean by "breeding environment". This section presents a lot of information. I wonder whether it would be more straightforward to make comparison between region not on a drug-by-drug basis but by comparing summary metrics of resistance that would help cutting through the complexity.

This assertion should be supported by numeric evidence to quantify the correlation "We also observed there was a strong correlation between the presence of ARGs and the expected resistance phenotypes"

I find the results section generally problematic; it reads very much like a report of the many experiments conducted. I present too much information including technical details that would be best suited for the supplementary material (ie "The 7.6_kb cassette was flanked by an IS6 and an IS3 elements, while the 2.4-kb cassette was flanked by a Tn3 and an IS6 elements. Sequence comparisons showed that the mcr-1-carrying plasmid pXD33-06 was highly homologous to plasmid pWI2-mcr (GenBank accession no. LT838201) (Figure 6C)". It lacks a structure, and a narrative that would encourage the read take more time to consider this section of the manuscript.

Methods:

I am not competent to judge the validity of the genetic analysis.

Discussion:

Line 137: what do the authors mean by "comparing from different sample collections" this need to be more strongly referenced in the main manuscript.

Reviewer #2 (Remarks to the Author):

This is an extensive and very complete epidemiological survey, focusing on antibiotic resistance among *E. coli* isolates recovered from pigs throughout mainland China. The collection of isolates is representative of most geographical area, and the design of the study is meaningful. Also, the methodology used is up-to-date, and the manuscript quite well written.

The manuscript would however benefit from significant shortening in many parts, and authors are encouraged to condense many parts of it.

The major limitation of this study is the complete lack of data with regard to the antibiotic consumption in those pig farms. This would have definitely add to the quality of the study, allowing to figure out which selective pressure was the most concerning.

The authors used WGS to decipher the resistance mechanisms of a large proportion of isolates, and this indeed provides interesting and complete data.

Page 11. Although most data in relation with resistance gene acquisition are meaningful and well presented, that part dealing with the so-called "point mutations" appears quite meaningless, and should probably be deleted. Indeed, there were mutations identified in many of those genes listed (by the way mutation should be rather considered as substitutions, clearly mentioning the reference genome), but their significance remains doubtful (in term of impact on the resistance phenotypes), or even meaningless if not clearly correlated with a resistance phenotype.

Page 13. It is not very clear whether some of those *mcr+* or *blaNDM+* plasmids share significant identities with plasmids carrying the same resistance genes encountered in human isolates. Please clarify.

The referee is surprised not to hear about possible novel resistance genes that would have been identified through such massive WGS approach. Do you confirm there was no new beta-lactamase gene identified ? same applies for other resistance genes....

Reviewer #3 (Remarks to the Author):

General comments

- Scale is impressive – good sampling spread
- Writing would benefit from a comprehensive English edit
- Evidence of inappropriate language "get its first inkling of impending catastrophe of a return to the pre-antibiotic."

Introduction

- Good coverage of the salient issues. However there is no mention of previous, bodies of work that describe whole genome sequences of *E. coli* from swine. There are several salient bodies of work that establish the dominant *E. coli* sequence types, particularly those that belong to phylogroup A clonal cluster 10 and the widespread carriage of class 1 integrons. This is significant omission because the authors have not put what is being found in *E. coli* in China in the context of *E. coli* in global swine production and failed to present what might be described as the most important information from their study data- that is a detailed description of *E. coli* phylogeny (sequence types and phylogroups) and pMLST and F plasmid replicon types.

Results

- What was the distribution by province? Were all farms represented in the end?
- Previous studies indicate that *E. coli* belonging to clonal complex 10 (CC10) predominate in swine. Is this contention supported or not in this study?
- Carriage of class 1 integrons in *E. coli* from healthy swine is significant. The authors might consider reporting carriage of the class 1 integrase gene *int11* in their *E. coli* population.
- Pigs carry high numbers of antibiotic resistant bacteria in the faeces even in countries that practice sound antimicrobial stewardship. What is needed is an in depth understanding of the AMR genes and their contexts and how they are mobilised (plasmids, genomic islands etc)

- Lines 114 onward. All those antimicrobials etc could be listed in methods and removed from results
- What is the split between healthy and diarrhoeal pigs? ST101 is a common ETEC isn't it?
- Selection for NGS could have been balanced by strains without those last-line antibiotic resistance phenotypes to better understand any similarity in the underlying genomic scaffolds
- VFGs seem to have a lot of ETEC genes - but there is no description about their distribution from healthy or diseased pigs. Are all the sampled pigs healthy?
- Point mutation section should be dramatically reduced in description to mainly describe the putative phenotypes. No need to say every AA change. Also it is important to relate the AA change to the antibiotic resistance invoked.
- Lines 254-257 – why are they only looking at plasmids across these genotypes? A thorough analysis of F plasmid MLST and replicon types should be reported
- No analysis of AMR genes or plasmids by phylogenetic background like ST or phylogroup. This is a significant omission
- Section: High genetic propensity of farm sourced XDR-E. coli in spreading into humans.
 - o Bioproject on line 315 doesn't exist – based on methods this is a typo and it should read "PRJNA400107" not "PRJNA4001047"
 - o SNPs over what alignment-define the parameters?
- What STs in the human collection?
 - o There is no interspecies transmission analysis. What evidence is presented that pig E. coli find their way into humans or at least are shared (<100 SNPs across a core)

Discussion

- Line 366 onwards: Consider comparing to collections from Chinese hospitals to raise impact? STs at least?
- Line 376 onward: No analysis of STs in conjunction with mcr genes – could have explored if the persistence of mcr in absence of colistin use is related to dominant STs that are co-selected by other factors?
- No discussion of STs in conjunction with virulence associated genes of certain pathotypes
- Line 433-435 - says many differed by less than 1000 and also less than 10 – results talk about less than 1000 and less than 100. Not consistent.

Methods

- Their main BioProject has not been released
- Line 526 – which ANI calculator?
- ParSNP was used for phylogenetic analysis
 - o This is not really appropriate for diverse genome collections
 - o No info on how ParSNP was used – what parameters?
 - o What was the size of the alignment?
 - o How many total SNPs?
 - o Why are the phylogenies generated by carriage of one gene?

Figures

- Presentation is inconsistent → →
- Figure 3a is problematic and it is difficult to determine which gene is which.
- All the trees in figure 4 look oddly similar but have different STs with the same colours – very confusing
 - o 4b - how are ST10 and ST48 so far apart given they reside in the same clonal complex?
 - o The tree is either mislabeled or the analyses have errors
 - o Presentation style is completely inconsistent
- Figure 7 describes 'Bayesian lineages' yet there is no mention of how they were determined in the methods
- Four figures of genomics data but very little descriptive results or discussion
- Overall, the genomic analysis needs to be significantly revised

Reviewer #4 (Remarks to the Author):

This is an interesting paper representing on AMR in *E. coli* from China and comprises a large body of work. It is worrying to see the high levels of resistance that was detected in the *E. coli* isolates purified from pigs and the farm environment, especially to last resort antibiotics. I have outlined below some specific comments/concerns on the paper:

- 1) The authors claim that the work represents pigs farms nationally but have only sampled 67 farms. It is important that they qualify what percentage of pig farms this number represents, and whether the distribution of farms sampled follows the national pattern i.e. more farms have been sampled from regions with higher number of pig farms. Or else they need to modify their statement.
- 2) They mention sampling pigs with diarrhoea in the Results section but no detail is provided in M+M. Presence of diarrhoea implies disease and details must be provided on how many pigs were healthy and how many showed disease. This data should then be correlated with presence of virulence genes, which has only been provided as a list with no context (In 185-192), which indicates lack interpretation/understanding. For example, presence of *stx2e* and *stb* genes in *E. coli* is often associated with diarrhoeal disease.
- 3) Very little detail is provided on the microbiology for purification of *E. coli* isolates. This should be provided in the M+M and referred to in the Results, where at least what type of agar was used for purification of *E. coli* should be mentioned and how many isolates were selected per farm.
- 4) The authors report 37% of isolates show phenotypic resistance to tigecycline, a last resort antibiotic. This is extremely high and it's important that a breakdown of genotype vs phenotype correlation is performed for these isolates. It is important to know if there's a particular MIC value which correlates with the plasmid mediated *tetX4* gene presence. In fact I would suggest that the authors perform a similar correlation between geno- and phenol-types in isolates harbouring all last resort or important antimicrobials.
- 5) I was disappointed with the work on phylogeny, this was not explored in any depth at all. I feel this could be much more informative than serotyping or MLST, and the least the authors could do is compare across these different methods and determine whether one can help inform on the other and whether pathogenic O-types/STs are clustering together.
- 6) List of the chromosomal SNPs without any correlation to associated phenotypes is meaningless (In 229-246). For example, for *gyrA*, only mutations in the QRDR region results in AMR phenotypes. The authors need to revise this section.
- 7) It's good to see the AMR plasmid data but providing Inc-types does not give any insight to which plasmid harbours the AMR gene, as isolates can have multiple plasmids. It would have been much better if the isolates for which closed plasmid genomes were available were used as reference to isolates in this dataset to see how similar plasmids were in other isolates bearing the same AMR gene (see Duggett et al, 2018).
- 8) The authors mention in the Discussion that presence of the AMR genes suggest extensive usage of different antimicrobials, however, it would be better if they could provide specific data. This could then help determine why only certain farms harbour particular AMR genes e.g. *tetX4*, if most farms may be using it (see Abuoun et al, 2020).

REVIEWER COMMENTS

Reviewer #1 (Remarks to the Author):

General comments

The study by Peng et al presents an overview of the trends in antimicrobial resistance in pigs China. It is a reasonably large survey with each province represented by approximately 50 isolates. The issue of AMR in animal production in China is important and deserves scrutiny. This study contributes to that objective, on an unprecedented geographical scale by combining phenotypic, and genetic assays. However, other studies have already shed light on this issue.

The study by Peng et al is potentially interesting, albeit challenges concerning the writing quality of the manuscript should be addressed before a more complete assessment of the work done can be conducted (see below). However, a key missing aspect of this study is a comparison with trends in AMR in animals reported by the National Surveillance program of the Ministry of Agriculture. This would considerably strengthen this study, and highlight potential divergences with this surveillance systems. I would also encourage the authors to support the statement made in the results and discussion with statistical test and quantitative evidence.

Response: We are grateful for the comments and suggestions on our manuscript. Yes, indeed, China has developed a National Surveillance program to monitor AMR in animals, and this project is hosted by the Ministry of Agriculture. When we were preparing the first version of this manuscript, we also took this idea into mind. However, the data of this National Surveillance program was not publicly accessible, and we did not have any channels to obtain these data. This is why we miss this part in our manuscript. Nevertheless, we have collected some published data from other articles or from the China Antimicrobial Resistance Surveillance System (CARSS, <http://www.carss.cn/>) and included these data in the discussion section as suggested. Please see lines 331-339.

The abstract is 380 words long instead of (maximum) 150.

The abstract contains a lot of elements of method where the reader would expect to see original results. This manuscript contains (too) many results, and (too) many figures. The authors should reconsider the amount and nature of information to be presented in the main manuscript to facilitate the work of the potential readers.

Response: Thank you for pointing this out for us. We have re-written the Abstract section and have made it meet the requirement. Also, we have re-organized our results section to make it clearer. Please see lines 103-262.

The written English of the manuscript is of poor quality, and makes the work difficult to follow. Example line 331: "To the best of our knowledge, this is the first time the AMR phenotypes of E. coli isolates from pig farms in all provinces of mainland China, including Tibet and Qinghai where there were little related data before being reported."

Response: We apologize for the language problems in the original manuscript. The language presentation was improved with the assistance from a native-speaker with appropriate research background. Those revised or rewritten places are marked in red in our revised submission. Thank you.

Results:

Could the authors specify what they mean by “breeding environment”. This section presents a lot of information. I wonder whether it would be more straightforward to make comparison between region not on a drug-by-drug basis but by comparing summary metrics of resistance that would help cutting through the complexity.

Response: We are sorry the inappropriate statement here. We mean samples collected from the “breeding environments” in pig farms (e.g., swabs of vehicle surface, fecal slurry, food troughs, nipple drinker, and floors). We have specified these sample types in the newly generated Figure 1A. Also, we have re-organized our result-presentation in this section. Please see lines 103-141. Thank you.

This assertion should be supported by numeric evidence to quantify the correlation “We also observed there was a strong correlation between the presence of ARGs and the expected resistance phenotypes”

Response: We are grateful for the suggestion. We have changed the statements here. Please see lines 178-179. Thank you.

I find the results section generally problematic; it reads very much like a report of the many experiments conducted. I present too much information including technical details that would be best suited for the supplementary material (ie “The 7.6_kb cassette was flanked by an IS6 and an IS3 elements, while the 2.4-kb cassette was flanked by a Tn3 and an IS6 elements. Sequence comparisons showed that the mcr-1-carrying plasmid pXD33-06 was highly homologous to plasmid pWI2-mcr (GenBank accession no. LT838201) (Figure 6C)”. It lacks a structure, and a narrative that would encourage the read take more time to consider this section of the manuscript.

Response: We agree with the comments and we have re-organized the result section. Please see lines 103-262. Thank you.

Methods:

I am not competent to judge the validity of the genetic analysis.

Discussion:

Line 137: what do the authors mean by “comparing from different sample collections” this need to

be more strongly referenced in the main manuscript.

Response: We apologize for the problems in the original manuscript. We have rewritten this part. Please see lines 112-124. Thank you.

Reviewer #2 (Remarks to the Author):

This is an extensive and very complete epidemiological survey, focusing on antibiotic resistance among E. coli isolates recovered from pigs throughout mainland China.

The collection of isolates is representative of most geographical area, and the design of the study is meaningful.

Also, the methodology used is up-to-date, and the manuscript quite well written.

Response: We are very grateful to your comments and suggestions for the manuscript. We have studied these constructive suggestions very carefully and have revised our manuscript accordingly. Blow please find our responses and modifications.

The manuscript would however benefit from significant shortening in many parts, and authors are encouraged to condense many parts of it.

The major limitation of this study is the complete lack of data with regard to the antibiotic consumption in those pig farms. This would have definitely added to the quality of the study, allowing to figure out which selective pressure was the most concerning.

Response: We are very grateful for this good suggestion. Although farmers at the majority of farms included for investigation did not want to disclose the antibiotics used at their facilities, **we still have got** some information from some farms and have added them into the revised manuscript (lines 295-301). In addition, several published data have also been added (lines 288-295). Thank you.

The authors used WGS to decipher the resistance mechanisms of a large proportion of isolates, and this indeed provides interesting and complete data.

Page 11. Although most data in relation with resistance gene acquisition are meaningful and well presented, that part dealing with the so-called "point mutations" appears quite meaningless, and should probably be deleted. Indeed, there were mutations identified in many of those genes listed (by the way mutation should be rather considered as substitutions, clearly mentioning the reference genome), but their significance remains doubtful (in term of impact on the resistance phenotypes), or even meaningless if not clearly correlated with a resistance phenotype.

Response: Thank you very much for this suggestion. We have removed this part in the revised manuscript.

Page 13. It is not very clear whether some of those mcr+ or blaNDM+ plasmids share significant identities with plasmids carrying the same resistance genes encountered in human isolates. Please

clarify.

Response: Thank you very much for this suggestion. Yes, these plasmids shared significant identities with plasmids carrying the same resistance genes encountered in human isolates. We have added the relative results in our revised manuscript. Please see lines 212-221, 225-231, and the newly generated figure 5.

The referee is surprised not to hear about possible novel resistance genes that would have been identified through such massive WGS approach. Do you confirm there was no new beta-lactamase gene identified? same applies for other resistance genes....

Response: Thank you very much for this comment. There are two popular ways to display the distribution of resistance genes across the strains: a binary matrix consisting of 1 for gene presence and 0 for absence, and a data matrix consisting of blast identity scores from 0 to 100%. Since there is no universal identity threshold value for estimation of the novelty of the resistance genes, a qualitative determination of gene presence and absence is chosen based on the resfinder package using the minimum identity of 90% and the minimum coverage of 60% in our manuscript. Based on these results, we do not find novel resistance genes right now.

Reviewer #3 (Remarks to the Author):

General comments

- *Scale is impressive – good sampling spread*
- *Writing would benefit from a comprehensive English edit*
- *Evidence of inappropriate language “get its first inkling of impending catastrophe of a return to the pre-antibiotic.”*

Response: We apologize for the language problems in the original manuscript. The language presentation was improved with the assistance from a native-speaker with appropriate research background. Those revised or rewritten places are marked in red in our revised submission. Thank you.

Introduction

- *Good coverage of the salient issues. However, there is no mention of previous, bodies of work that describe whole genome sequences of E. coli from swine. There are several salient bodies of work that establish the dominant E. coli sequence types, particularly those that belong to phylogroup A clonal cluster 10 and the widespread carriage of class 1 integrons. This is significant omission because the authors have not put what is being found in E. coli in China in the context of E. coli in global swine production and failed to present what might be described as the most important information from their study data- that is a detailed description of E. coli phylogeny (sequence types and phylogroups) and pMLST and F plasmid replicon types.*

Response: We are grateful for the suggestion. We have added the suggested contents (“Recently, analyzing E. coli isolates from pigs and other animals by antimicrobial susceptibility testing (AST)

in combination with whole genome sequencing (WGS) facilitates a better understanding of AMR development and dissemination. For example, a recent study has found a plasmid-borne *mcr-1* is responsible for the rapid dissemination of polymyxin resistance in Enterobacteriaceae in animals and humans. Another recent study has revealed that IncX4, IncI2, and IncHI2 are the common plasmid types carrying *mcr-1*-positive *E. coli* in pig farms after the cessation of colistin use as a feed additive for animals in China, and significant increases in IncI2-associated *mcr-1* and a distinct lineage of *mcr-1*-associated IncHI2 are observed post ban. Using WGS data, many studies have also reported that *E. coli* sequence type (ST) 10 and its related types are important AMR-associated genotypes predominate in swine and a higher prevalence of plasmid-borne ARGs is found in these types compared to others.”) in the Introduction section. Please see lines 70-86. Thank you.

Results

- *What was the distribution by province? Were all farms represented in the end?*

Response: In each of the 31 provinces in mainland China, 2~3 different pig farms were randomly selected to collect swabs from fresh feces and rectal swabs of pigs (with approximately 40 samples per farm), as well as swabs of drinking and fecal slurry, floors, troughs (for each point at least three samples per farm were collected) (lines 416-420). We have rewritten this part and make it clearer. Please see lines 112-141. As we mentioned in our manuscript (lines 108-111, 393-395), the sudden outbreak of African Swine Fever (ASF) in late 2018 and COVID-19 in late 2019 made it extremely difficult to get more samples from more farms. Therefore, the project was briefly interrupted. However, we will not stop here and we will continuously monitor the distribution of drug-resistant bacteria in livestock for long time, we will re-start the sample collection when the conditions of ASF and COVID-19 get better. Thank you.

- *Previous studies indicate that *E. coli* belonging to clonal complex 10 (CC10) predominate in swine. Is this contention supported or not in this study?*

Response: Yes, it is supported and we have specified this point in the result section (lines 154-156) and the discussion section (lines 351-360). Thank you.

- *Carriage of class 1 integrons in *E. coli* from healthy swine is significant. The authors might consider reporting carriage of the class 1 integrase gene *intI1* in their *E. coli* population.*

Response: We are grateful to this good suggestion. We have done the analysis and have added the data in the result section (lines 185-190, supplementary materials Table S4) and the discussion section (lines 363-366). Thank you.

- *Pigs carry high numbers of antibiotic resistant bacteria in the faeces even in countries that practice sound antimicrobial stewardship. What is needed is an in depth understanding of the AMR genes and their contexts and how they are mobilised (plasmids, genomic islands etc)*

Response: We are grateful to this good suggestion which could improve the depth of our work. We have analyzed the carriage of plasmids and the class 1 integrons in our sequenced *E. coli*. Please see

lines 185-244 in the result section and lines 364-384 in the discussion section, figures 3~5, supplementary materials figures S2 and S3, Table S4, Table S5, and Table S6.

• *Lines 114 onward. All those antimicrobials etc could be listed in methods and removed from results*

Response: We have removed this part from the result section and have listed it in methods (lines 104-124; lines 433-441). Thank you for the suggestion.

• *What is the split between healthy and diarrhoeal pigs? ST101 is a common ETEC isn't it?*

Response: Thank you for the comments. Both ST10, ST101, and ST48 have been determined in *E. coli* pathotypes (ETEC/EPEC/STEC) from pigs with diarrhea (PMID: 31480723). In our study, these three STs were also commonly determined among the 207 isolates from the feces of diarrheal pigs [ST10 ($n = 28$), ST101 ($n = 14$), ST48 ($n = 8$)] (lines 355-360).

• *Selection for NGS could have been balanced by strains without those last-line antibiotic resistance phenotypes to better understand any similarity in the underlying genomic scaffolds*

Response: Thank you for the comments. We have rewritten this part. Please see lines 144-146. The NGS was not only performed on those isolates with resistance phenotypes to the last-line antibiotics, but also on those isolates with resistance phenotypes to broad-spectrum-cephalosporins ($n = 495$), aminoglycosides ($n = 334$), phenicols ($n = 476$), tetracyclines ($n = 510$), fluoroquinolones ($n = 473$), sulfonamides ($n = 452$), and/or nitrofurantoin ($n = 59$) (Supplementary materials Table S1).

• *VFGs seem to have a lot of ETEC genes - but there is no description about their distribution from healthy or diseased pigs. Are all the sampled pigs healthy?*

Response: We are very grateful to your comments and suggestion. We have specified this content (“only 36.64% (66/181) of these isolates were recovered from the feces of diarrheal pigs, and the remaining were from environmental samples. The wide presence of these MDR-*E. coli* in pig farms represents a great health risk and should receive more attentions.”). Please see lines 348-350. Thank you.

• *Point mutation section should be dramatically reduced in description to mainly describe the putative phenotypes. No need to say every AA change. Also it is important to relate the AA change to the antibiotic resistance invoked.*

Response: Thank you very much for this comment. We have removed this part in the revised manuscript according to the suggestion from the other reviewers.

• *Lines 254-257 – why are they only looking at plasmids across these genotypes? A though analysis of F plasmid MLST and replicon types should be reported*

Response: Thank you for your comments. Actually, we have analyzed the plasmid replicons and the sequence types of all the 515 sequenced MDR-*E. coli* (Tables S1 and S5 in supplementary materials, lines 191-193). During the analysis we found that three types of plasmids, including ColR_{NAI} (n = 272), IncF_{IB} (n = 266), and IncX₁ (n = 236) were presented in most of the isolates (lines 193-195; Supplementary materials Figure S2A; Table S5). After check, we also found these three types of plasmids were carried by most of the ESBL-genes carrying isolates [ColR_{NAI} (260/495), IncF_{IB} (257/495), IncX₁ (228/495)], quinolone-resistant isolates [ColR_{NAI} (248/473), IncF_{IB} (257/473), IncX₁ (228/473)], and tetracycline-resistant isolates [ColR_{NAI} (270/510), IncF_{IB} (265/510), IncX₁ (235/510)] (lines 195-199; Supplementary materials Table S5). However, these ESBL-genes carrying isolates, quinolone-resistant isolates, and tetracycline-resistant isolates were also resistant to many other antimicrobial classes tested (Supplementary materials Table S1). The sequence types of these isolates are also given in Table S1 in supplementary materials.

• *No analysis of AMR genes or plasmids by phylogenetic background like ST or phylogroup. This is a significant omission*

Response: Thank you for your comments. We have put the required information in the newly generated Table S1 in supplementary materials. We did not put these results in the main text because the limitation of the length of the manuscript. In addition, we have also showed the results in the newly generated Figure 3 and Figure 4.

• *Section: High genetic propensity of farm sourced XDR-*E. coli* in spreading into humans.*
o Bioproject on line 315 doesn't exist – based on methods this is a typo and it should read “PRJNA400107” not “PRJNA4001047”

Response: We apologize for our carelessness. We have revised the Bioproject number. It is indeed “PRJNA400107”. Thank you.

o SNPs over what alignment-define the parameters?

What STs in the human collection?

*o There is no interspecies transmission analysis. What evidence is presented that pig *E. coli* find their way into humans or at least are shared (<100 SNPs across a core)*

Response: Thank you very much for your comments. SNPs were detected and exported using Parsnp v1.5.3 with the options -p 36 -c -C 100 -z 25 (Lines 486-488). STs of the human isolates were added and were presented in the result section (lines 257-262) and were shown in Figure 6B as well as in supplementary materials Table S8. From the phylogenetic and genomic level, the genome sequences of the pig farm originated MDR-*E. coli* were very close to the human *E. coli* (Figure 6A and Table S7). In addition, many STs including the predominantly determined STs in this study (e.g., ST10, ST48, ST206, ST101, etc.) were shared by the farm originated *E. coli* and the human isolates (Figure 6B). Therefore, we speculated that a very high genetic propensity of farm sourced MDR-*E. coli* in spreading into humans. Similar tests and speculation have been made and stated in the other publications (e.g., PMID: 32737421)

Discussion

• *Line 366 onwards: Consider comparing to collections from Chinese hospitals to raise impact? STs at least?*

Response: Thank you for your suggestion. We have analyzed and discussed the STs shared by the human isolates and the pig farm originated E. coli isolates (lines 257-262, 359-360, 386-387, Figure 6B).

• *Line 376 onward: No analysis of STs in conjunction with mcr genes – could have explored if the persistence of mcr in absence of colistin use is related to dominant STs that are co-selected by other factors?*

Response: Thank you for your comments. Analysis of STs in conjunction with mcr genes is shown in Figure 4C. Also, we have re-written the content about the colistin-resistant isolates in the discussion section (lines 308-311; “This relatively low proportion of recovery is attributed to China’s policy to ban the use of colistin in agriculture in 2017, which has had a significant effect on reducing colistin resistance in both animals and humans”).

• *No discussion of STs in conjunction with virulence associated genes of certain pathotypes*

Response: Thank you for your comments. The related contents have been added (lines 345-360). Detailed results are also given in supplementary materials Table S2.

• *Line 433-435 - says many differed by less than 1000 and also less than 10 – results talk about less than 1000 and less than 100. Not consistent.*

Response: Thank you for pointing this out and sorry for our carelessness. We have revised the statement and made them in consistence (line 389).

Methods

• *Their main BioProject has not been released*

Response: Thank you for pointing this out for us. We have asked NCBI staff to release the sequences determined in this study. Blow I attached their respond to our request email:

[REDACTED]

[REDACTED]

- *Line 526 – which ANI calculator?*

Response: This ANI calculator (<http://enve-omics.ce.gatech.edu/ani/>) was developed by Kostas Lab at Georgia Institute of Technology. We have added the weblink in our revised manuscript (line 484). Thank you.

- *ParSNP was used for phylogenetic analysis*
 - o *This is not really appropriate for diverse genome collections*
 - o *No info on how ParSNP was used – what parameters?*
 - o *What was the size of the alignment?*
 - o *How many total SNPs?*
 - o *Why are the phylogenies generated by carriage of one gene?*

Response: Parsnp v1.5.3 was used for detection of core-genome SNPs and reconstruction of phylogenetic trees with the options -p 36 -c -C 100 -z 25 (Lines 486-488). The core-genome multiple alignments consisted of locally collinear blocks (> 25 bp). The complete genome sequence of the strain XD35 was used as reference for the phylogenetic analyses. The details of the core-genome SNP alignments have been summarized for the phylogenies of the strains belonging to the individual antibiotic class.

Types of antibiotics	No. of genomes	Size of SNP alignment (bp)
Carbapenem	49	135971
Colistin	71	162734
Quinolone	473	154360

Tetracycline	510	150961
ESBL	495	77384
515pigs + 287human	802	49408

Figures

- *Presentation is inconsistent*

Response: Thank you for pointing this for us. We have regenerated the figures and made them clear.

- *Figure 3a is problematic and it is difficult determine which gene is which.*

Response: We have removed Figure 3a and have put the displayed information in supplementary materials Table S1 to make it easy to be readable. Also, we have regenerated all figures and made them easy to read.

- *All the trees in figure 4 look oddly similar but have different STs with the same colours – very confusing*
 - o *4b - how are ST10 and ST48 so far apart given they reside in the same clonal complex?*
 - o *The tree is either mislabeled or the analyses have errors*
 - o *Presentation style is completely inconsistent*

Response: The colors of STs have been revised in Figure 4 and each ST is consistent with the corresponding color scheme. Also, we divided this figure into two figures (Figures 3&4) in the revised manuscript to make them easy to read.

- *Figure 7 describes 'Bayesian lineages' yet there is no mention of how they were determined in the methods*
- *Four figures of genomics data but very little descriptive results or discussion*
- *Overall, the genomic analysis needs to significantly revised*

Response: We are grateful to your comments and suggestions. We have removed Figure 7a and regenerated all figures in the revised manuscript to make them easy to read. We have also rewritten the results and discussion section according to your suggestion. Please see the rewritten part in our revised manuscript. Thank you.

Reviewer #4 (Remarks to the Author):

This is an interesting paper representing on AMR in E. coli from China and comprises a large body of work. It is worrying to see the high levels of resistance that was detected in the E. coli isolates purified from pigs and the farm environment, especially to last resort antibiotics. I have outlined below some specific comments/concerns on the paper:

- 1) *The authors claim that the work represents pigs' farms nationally but have only sampled 67 farms. It is important that they qualify what percentage of pig farms this number represents, and whether*

the distribution of farms sampled follows the national pattern i.e., more farms have been sampled from regions with higher number of pig farms. Or else they need to modify their statement.

Response: Yes, the distribution of farms sampled follows the national pattern (we collected more samples in those provinces with more pigs rearing). We have rewritten this part and make it clearer (lines 108-111, 393-395), the sudden outbreak of African Swine Fever (ASF) in late 2018 and COVID-19 in late 2019 made it extremely difficult to get more samples from more farms. Therefore, the project was briefly interrupted. However, we will continuously monitor the distribution of drug-resistant bacteria in livestock due to its high impact, we will restart the sample collection when the conditions of ASF and COVID-19 get better. Thank you.

2) They mention sampling pigs with diarrhoea in the Results section but no detail is provided in M+M. Presence of diarrhoea implies disease and details must be provided on how many pigs were healthy and how many showed diseases. This data should then be correlated with presence of virulence genes, which has only been provided as a list with no context (ln 185-192), which indicates lack interpretation/understanding. For example, presence of stx2e and stb genes in E. coli is often associated with diarrhoeal disease.

Response: Thank you for your comments and suggestions. Numbers of isolates from diarrheal pigs and the other sample types have been added and displayed in the newly generated Figure 1a. Detailed results about the presence of O-serogroups, STs, and VFGs have been listed in the regenerated Table S2 in the supplementary materials. The suggested contents (e.g., presence of stx2e and stb genes in E. coli is often associated with diarrhoeal disease and the other contents) have been discussed at lines 355-360 in the discussion part.

3) Very little detail is provided on the microbiology for purification of E. coli isolates. This should be provided in the M+M and referred to in the Results, where at least what type of agar was used for purification of E. coli should be mentioned and how many isolates were selected per farm.

Response: Thank you for your comments and suggestions. Please find the revised statement at lines 423-429.

4) The authors report 37% of isolates show phenotypic resistance to tigecycline, a last resort antibiotic. This is extremely high and it's important that a breakdown of genotype vs phenotype correlation is performed for these isolates. It is important to know if there's a particular MIC value which correlates with the plasmid mediated tetX4 gene presence. In fact, I would suggest that the authors perform a similar correlation between geno- and phenol-types in isolates harbouring all last resort or important antimicrobials.

Response: We are grateful to this comment. “Although a large number of E. coli isolates from pig farms in China showed resistance to tigecycline (37.31%, n = 698), most of them displayed low-level of resistance (MIC values ranging from 0.5 µg/ml to 1 µg/ml; 92.98%, n = 649), and only five isolates displayed high-level of resistance (MIC ≥ 4 µg/ml; 0.72%). A high proportion of recovering isolates with low-level of tigecycline resistance may be because the breakpoint used for the

interpretation of tigecycline resistance recommended by the European Committee on Antimicrobial Susceptibility Testing (EUCAST) is low (0.5 µg/ml, EUCAST document v 8.1). The phenotype of low-level of tigecycline resistance may be conferred by tetracycline-resistant genes (e.g., tetA, tetB, tetC, and/or tetM) since those genes were commonly harbored by *E. coli* from animals.” We have added the explanations in the discussion section (lines 311-321). Also, in the result section, we have shown the correlation between geno- and phenol-types in isolates harbouring all last resort or important antimicrobials (lines 178-182).

5) I was disappointed with the work on phylogeny, this was not explored in any depth at all. I feel this could be much more informative than serotyping or MLST, and the least the authors could do is compare across these different methods and determine whether one can help inform on the other and whether pathogenic O-types/STs are clustering together.

Response: Thank you very much for your comments. Since the phylogenetic tree shown in Figure 3a is not easy to read, we have removed this figure and put all the information in the newly generated supplementary materials Table S1 and Table S2 according to the suggestion of some other referees. In addition, we regenerated Figures 3 and 4 in our revised manuscript and the results shown in these two figures are clear and easy to read.

6) List of the chromosomal SNPs without any correlation to associated phenotypes is meaningless (ln 229-246). For example, for gyrA, only mutations in the QRDR region results in AMR phenotypes. The authors need to revise this section.

Response: Thank you very much for this suggestion. We have removed this part in the revised manuscript according to the suggestion of some other referees.

7) It's good to see the AMR plasmid data but providing Inc-types does not give any insight to which plasmid harbours the AMR gene, as isolates can have multiple plasmids. It would have been much better if the isolates for which closed plasmid genomes were available were used as reference to isolates in this dataset to see how similar plasmids were in other isolates bearing the same AMR gene (see Duggett et al, 2018).

Response: We are grateful to this useful suggestion and the good example you provided. We have re-performed the analysis and put the results in the newly generated Figure 5 as well as in supplementary materials figures S2&S3 in the revised manuscript following your suggestion.

8) The authors mention in the Discussion that presence of the AMR genes suggest extensive usage of different antimicrobials, however, it would be better if they could provide specific data. This could then help determine why only certain farms harbour particular AMR genes e.g. tetX4, if most farms may be using it (see Abuoun et al, 2020).

Response: Thank you very much for providing us this useful suggestion and the good example published by Abuoun. After we received your comments, we conducted a questionnaire investigation on the usage of antimicrobial agents in the pig farms included in this study. However,

farmers at the majority of farms included for investigation did not hope to disclose the antibiotics used at their facilities, several farmers disclosed that antibiotics belonging to sulphonamides (e.g., sulfamonomethoxine), tetracyclines (e.g., chlortetracycline, doxycycline, oxytetracycline), fluoroquinolones (e.g., enrofloxacin), macrolides (e.g., ivermectin, kitasamycin tartrate), and β -lactams (e.g., cefotaxime, penicillin, amoxicillin) were used in their farms for growth promotion or disease therapy. We have included this information and have re-written the parts in the discussion section (lines 284-301), but current limited data might be not enough to link the presence of the AMR genes and the extensive usage of different antimicrobials in pig farms.

Thank you again for your kind comments.

REVIEWER COMMENTS

Reviewer #2 (Remarks to the Author):

My comments have been adequately considered, and I congratulate the authors for all the efforts they did to address all queries.

Reviewer #3 (Remarks to the Author):

Overall comments.

While the manuscript has benefited for a first round of review, there are still many areas that require improvement. The current iteration of the abstract hardly conveys novel information. Conclusions drawn are not substantiated because inappropriate analyses have been used. A

The concluding sentence of the abstract "Our data presented herein will help understand the current AMR profiles in pigs and also provide reference for policy formulation of AMR control action in livestock in China." is diminished significantly because the bioinformatic methods are still vague and inadequate with obvious omissions.

There are no analysis scripts available to reproduce figures and analysis from the raw data. Some figures have clearly been generated in R with ggtree and other software yet there is no mention of this in the methods. R version and package versions need to be provided. Analysis (generation of all statistics and figures) for a paper of this scale in a journal of this standard should be performed programmatically, with raw data and scripts provided so that a reader can download these materials and reproduce everything with ease. Raw data availability is not sufficient in our view. Specific points

There is no description of methods for the tree in Figure 2b, which appears to be generated via Enterobase. Enterobase needs to be cited and the tree-building method needs to be described and cited. The authors should consider seeking outside help with bioinformatics since several of the important items outlined in the first review remain unaddressed and are not up to standard.

There are significant issues with all the phylogenetic analyses.

Firstly, the methodology is inappropriate, and the results are therefore dubious. As stated in the first round of review, ParSNP is not appropriate for inferring phylogenies of diverse genomes. It is designed for very closely related genomes such as those from the same sequence type or from an outbreak (see: <https://harvest.readthedocs.io/en/latest/content/parsnp/faq.html>).

Its limitation is clear in Figures 3 and 4 where ST10 appears to be polyphyletic. In a diverse collection, individual STs should be monophyletic. Figure 6A is also rendered unreliable. We previously noted that CC10 and particularly ST10 is a dominant sequence type in pigs globally. The authors should acknowledge prior *Escherichia coli* genomic studies in swine (PMID: 29306352) with special attention to clonal complex 10 (PMID: 30303480).

Stemming from this is the issue of SNP distances. The revised manuscript still does not report the sizes of core genome alignments, or what percentage of a typical 5Mb genome they cover. This renders comments about 'closely related isolates' at distances of 10-1000 SNPs completely irrelevant.

Secondly, there is no justification for performing phylogenetic analysis by resistance phenotype because these phenotypes can be conferred by diverse resistance genes, which are typically independently mobile and therefore fundamentally detached from core genome phylogeny (except for cases of mass clonal expansion, like ST131 Clade C). It would be better to simply analyse presence/absence of genes for the respective phenotypes and determine if they are associated with any specific plasmid types. If any trends arise from that analysis, then ST-based trends could be analysed, however the trees as they are currently presented add nothing, particularly

given how unreliable they are.

The analysis in Figure 6A is also inadequate. It is well known that *E. coli* from different sources are dispersed across the species tree so it is not at all surprising that human and pig isolates are present close to one another on this tree. However, as mentioned, due to the inappropriate phylogenetic analyses there is no real resolution. STs exhibit subpopulation structure, so identifying the same ST in humans and pigs is no clear indication of transfer between these sources, especially when the phylogenetic methods are inappropriate.

The authors should regenerate the tree in figure 6A either with one of the simple Enterobase-based tree generation methods and note its limitations or with Snippy, rooting the phylogeny on an outgroup sequence of *Shigella*. If Snippy is used, the size of the full alignment and number of core SNPs should be reported. Again, please seek collaborative assistance from experienced people in the area.

As suggested by Reviewer 4, the authors need to use the complete plasmid sequences they generated as reference sequences and align their short reads from the pig collection to them to infer the presence of these plasmids in pigs. Alignment to a few random plasmids from GenBank is not informative. The paper could be greatly improved by inferring how many isolates carry these plasmids with last line AMR genes.

Inconsistency in figures 4b and 4c having text labels for STs instead of colours shown in other figures.

Figures 5a-c have no legend for the colour coding of genes.

The legend for 5d, 5e are not clearly split for each plasmid.

Please discuss PMID: 29306352 and PMID: 30303480 as they cover phylogenetics and integrons in swine and ST10 in swine respectively.

Minor comments:

Please see the appended word file. It contains a large number of suggested edits and minor typographic errors.

Reviewer #4 (Remarks to the Author):

The manuscript is much improved and I felt the extra work that the authors have performed has really helped to understand the data better, in particular the plasmid aspects. However, I felt there were still some outstanding points worth considering:

- 1) The grammar needs to be improved/revised in places.
- 2) Line 138 – I think "high" was missing after relatively?
- 3) Line 155 – not sure what "broadly determined" means do authors mean most prevalent?
- 4) Line 252-257 (Results) and Line 387-390 (Discussion) - I strongly disagree that 1000 SNP differences or even a 100 SNP differences between isolates are close. In my opinion something more like <20 SNPs are close. I suggest that the authors add a reference and justification for this claim, or revise this statement.
- 5) Line 311-318 – Mostly repeat of what is already in the Results – suggest that it is revised/shortened.
- 6) Line 322-344 – Plasmid mediated carbapenem resistance was detected on farms although it is banned from usage in China. Although the authors have provided two possible explanation, a third possibility worth considering is that the plasmid and/or isolate was co-selected by use of another antimicrobial that may be present in the plasmid and/or isolate.

REVIEWER COMMENTS

Reviewer #2 (Remarks to the Author):

My comments have been adequately considered, and I congratulate the authors for all the efforts they did to address all queries.

We sincerely acknowledge your contributions to the review of our manuscript. Your constructive suggestions and comments have helped us improve the presence of our work clearly. Thank you very much again.

Reviewer #3 (Remarks to the Author):

Overall comments.

While the manuscript has benefited for a first round of review, there are still many areas that require improvement. The current iteration of the abstract hardly conveys novel information. Conclusions drawn are not substantiated because inappropriate analyses have been used.

We are grateful to your comments and patience on our manuscript. Regarding one of your main concerns which is about the phylogenetic analysis, **we have reperformed all analyses using the tool “Snippy” you recommended. Two excellent works you mentioned (PMID: 29306352 & PMID: 30303480) have been also acknowledged in this version of our manuscript. For the language problem, we have asked a service from Nature Research Editing Service.** Below please find our response point-by-point:

The concluding sentence of the abstract “Our data presented herein will help understand the current AMR profiles in pigs and also provide reference for policy formulation of AMR control action in livestock in China.” is diminished significantly because the bioinformatic methods are still vague and inadequate with obvious omissions.

Thank you for your comments. **We have reperformed the analysis based on the tool according to your recommendation (using “Snippy”). For the other parts, we have also performed analyses according to your suggestions.** Please checked all the revised parts we mentioned below again in their corresponding places in the revised manuscript.

There are no analysis scripts available to reproduce figures and analysis from the raw data. Some figures have clearly been generated in R with ggtree and other software yet there is no mention of this in the methods. R version and package versions need to be

provided. Analysis (generation of all statistics and figures) for a paper of this scale in a journal of this standard should be performed programmatically, with raw data and scripts provided so that a reader can download these materials and reproduce everything with ease. Raw data availability is not sufficient in our view.

We have provided more details in the Materials and methods section. “Phylogenetic trees and the related metadata were visualized by using the package `ggtree` v3.2.0 (PMID: 32162851) in the R version 4.1.0 environment” (lines 486-500). **R scripts were provided in Text 1 in supplementary materials (lines 498-500, 545-546).** Thank you.

Specific points

There is no description of methods for the tree in Figure 2b, which appears to be generated via Enterobase. Enterobase needs to be cited and the tree-building method needs to be described and cited. The authors should consider seeking outside help with bioinformatics since several of the important items outlined in the first review remain unaddressed and are not up to standard.

Figure 2B was generated according to the following method: “The sequence type of each genome was detected based on the program MLST (<https://github.com/tseemann/mlst>) incorporating components of the PubMLST database (PMID: 21143983). The minimal spanning tree of the sequenced *E. coli* isolates was constructed using GrapeTree version 1.5.0 (PMID: 30049790), an interactive tree visualization program within Enterobase.” **We have added the methods in the materials and methods section (lines: 483-486).** Please check it. Thank you.

There are significant issues with all the phylogenetic analyses.

Firstly, the methodology is inappropriate, and the results are therefore dubious. As stated in the first round of review, ParSNP is not appropriate for inferring phylogenies of diverse genomes. It is designed for very closely related genomes such as those from the same sequence type or from an outbreak (see: <https://harvest.readthedocs.io/en/latest/content/parsnp/faq.html>).

Thank you for pointing this out for us. **We have reformed the phylogenetic analyses using Snippy**, a tool you recommended below. Please checked lines

*Its limitation is clear in Figures 3 and 4 where ST10 appears to be polyphyletic. In a diverse collection, individual STs should be monophyletic. Figure 6A is also rendered unreliable. We previously noted that CC10 and particularly ST10 is a dominant sequence type in pigs globally. The authors should acknowledge prior *Escherichia coli* genomic studies in swine (PMID: 29306352) with special attention to clonal complex*

10 (PMID: 30303480).

Thank you very much for the comments. Figures 3, 4, and 6 were generated based on the core-genome SNPs. In many studies, including the two articles *PMID: 29306352* and *PMID: 30303480*, isolates belonging to the same STs were polyphyletic (please see below the two figures adopted from *PMID: 29306352* and *PMID: 30303480*). **Therefore, in the revised manuscript we have generated a phylogenetic tree based on the MLST data and in this tree, and isolates belonging to the same STs were monophyletic (see the newly generated Figure 3).** According to your below suggestion, we used this newly figure to display the results of “the presence/absence of genes for the respective phenotypes and their association with the specific plasmid types”. **Also, we have used the recommended tool “Snippy” to regenerate Figure 6A as suggested below (please check the newly generated Figure 5 in the revised manuscript). Details about the two trees are also provided as suggested** (Please also check below their exact places in the revised manuscript according to our responses to your below comments). We have also acknowledged the two articles in the revised manuscript according to your suggestions (**lines 86, 354-356, 361-363, references 24 and 25**).

PS:

In *PMID: 29306352*, isolates belonging to the same STs (e.g., the ST10 isolates) were polyphyletic.

[REDACTED]

Fig.1. A mid-point rooted, maximum-likelihood phylogenetic tree inferred using PhyloSift v1.0.1, FastTree2, FigTree v1.4.2 and iTOL. (Adopted from Figure 1 in *PMID: 29306352*).

In *PMID: 30303480*, isolates belonging to the same STs (e.g., the ST10 isolates) were also polyphyletic.

[REDACTED]

Fig. 1. Maximum-likelihood phylogeny of 248 *E. coli* CC10 sequences. (Adopted from Figure 1 in *PMID: 30303480*).

Stemming from this is the issue of SNP distances. The revised manuscript still does not report the sizes of core genome alignments, or what percentage of a typical 5Mb genome they cover. This renders comments about ‘closely related isolates’ at distances of 10-1000 SNPs completely irrelevant.

For the core genome alignments of individual isolates, **a table showing the percentage of the XD35 genome (GenBank accession no. CP089142) as reference they cover was attached in Table S9 in supplementary materials. Please check lines 496-497, 542-543, and Table S9. Thank you.**

Secondly, there is no justification for performing phylogenetic analysis by resistance phenotype because these phenotypes can be conferred by diverse resistance genes, which are typically independently mobile and therefore fundamentally detached from core genome phylogeny (except for cases of mass clonal expansion, like ST131 Clade C). It would be better to simply analyse presence/absence of genes for the respective phenotypes and determine if they are associated with the any specific plasmid types. If any trends arise from that analysis, then ST-based trends could be analysed, however

the trees as they are currently presented add nothing, particularly given how unreliable they are.

We are grateful to this constructive suggestion. **We have reperformed the analysis according to your suggestions** (“simply analyse presence/absence of genes for the respective phenotypes and determine if they are associated with the any specific plasmid types. If any trends arise from that analysis, then ST-based trends could be analysed”) **and have displayed the results in the newly generated Figure 3**. Please see **lines 167-183 and the newly generated Figure 3** in the revised manuscript.

The analysis in Figure 6A is also inadequate. It is well known that E. coli from different sources are dispersed across the species tree so it is not at all surprising that human and pig isolates are present close to one another on this tree. However, as mentioned, due to the inappropriate phylogenetic analyses there is no real resolution. STs exhibit subpopulation structure, so identifying the same ST in humans and pigs is no clear indication of transfer between these sources, especially when the phylogenetic methods are inappropriate.

First, we have reperformed the analysis and regenerated the figure using the tool you recommended (“Snippy”). Please check the newly generated Figure 5 and lines 490-497 in the revised manuscript.

Secondly, we displayed the sequence types of both pig and human isolates here in a Venn diagram according to your comments (*What STs in the human collection?*) in the first round of review. Thank you.

The authors should regenerate the tree in figure 6A either with one of the simple Enterobase-based tree generation methods and note its limitations or with Snippy, rooting the phylogeny on an outgroup sequence of Shigella. If Snippy is used, the size of the full alignment and number of core SNPs should be reported. Again, please seek collaborative assistance from experienced people in the area.

Thank you for your comments. **We have reperformed the analysis and regenerated the figure using the tool you recommended (“Snippy”). Please check the newly generated Figure 5 and lines 490-497** in the revised manuscript.

“A core SNP alignment was generated for the 515 porcine strains, 287 human strains, and an *Escherichia fergusonii* strain by using the snippy-multi program (the strain XD35 used as reference) implemented in Snippy v 4.4.0. The size of the full alignment is 5,434,282 bp in length and the number of core SNPs is 170,888”.

As suggested by Reviewer 4, the authors need to use the complete plasmid sequences they generated as reference sequences and align their short reads from the pig collection to them to infer the presence of these plasmids in pigs. Alignment to a few

random plasmids from GenBank is not informative. The paper could be greatly improved by inferring how many isolates carry these plasmids with last line AMR genes.

Thank you for your comments. Indeed, we have performed the analysis using the complete plasmid sequences they generated as reference sequences and align their short reads from the pig collection to them to infer the presence of these plasmids in pigs. We have presented the results in **lines 211-213** in the result section as well as in **Figure S3**). However, we also aligned the complete genome sequences of the plasmids to those of the plasmids from GenBank because in the first round of review, Reviewer 2 asked us to clarify whether some of those *mcr+* or *blaNDM+* plasmids share significant identities with plasmids carrying the same resistance genes encountered in human isolates (“*It is not very clear whether some of those mcr+ or blaNDM+ plasmids share significant identities with plasmids carrying the same resistance genes encountered in human isolates. Please clarify.*”). Therefore, we also aligned the sequences of plasmids we generated with those from humans isolates in GenBank.

Inconsistency in figures 4b and 4c having text labels for STs instead of colours shown in other figures.

According to your suggestion, we have reformed the analysis and regenerated the figure. The originated Figure 4 has been deleted in this revised version (**please check the newly generated Figure 3**). Thank you.

Figures 5a-c have no legend for the colour coding of genes.

According to your suggestion, we have reformed the analysis and regenerated the figure. The originated Figure 5 has been deleted in this revised version (**please check the newly generated Figure 3**). Thank you.

The legend for 5d, 5e are not clearly split for each plasmid.

According to your suggestion, we have reformed the analysis and regenerated the figure. The originated Figure 5 and its legends have been deleted in this revised version. Thank you.

Please discuss PMID: 29306352 and PMID: 30303480 as they cover phylogenetics and integrons in swine and ST10 in swine respectively.

We have also acknowledged the two articles in the revised manuscript in the discussion section according to your suggestions (lines 354-356, 361-363, references 24 and 25). Thank you for this useful suggestion.

Minor comments:

Please see the appended word file. It contains a large number of suggested edits and minor typographic errors.

Thank you for your comments and help. We have requested a language service from Nature Research Editing Service (https://authorservices.springernature.com/go/nr/?utm_source=nroasLetters&utm_medium=email&utm_campaign=natcommsletters). Below please find a certificate for the Edition.

[REDACTED]

Once again, thank you very much for your comments and suggestions.

Reviewer #4 (*Remarks to the Author*):

The manuscript is much improved and I felt the extra work that the authors have performed has really helped to understand the data better, in particular the plasmid aspects. However, I felt there were still some outstanding points worth considering:

1) The grammar needs to be improved/revised in places.

Thank you for your comments and help. We have requested a language service from Nature Research Editing Service (https://authorservices.springernature.com/go/nr/?utm_source=nroasLetters&utm_medium=email&utm_campaign=natcommsletters). Below please find a certificate for the

Edition. Below please find a certificate for the Edition.

[REDACTED]

2) Line 138 – I think "high" was missing after relatively?

We have revised this place, please see line 137. Thank you very much.

3) Line 155 – not sure what "broadly determined" means do authors mean most prevalent?

We have changed the state, please see line 155. Thank you very much.

4) Line 252-257 (Results) and Line 387-390 (Discussion) - I strongly disagree that 1000 SNP differences or even a 100 SNP differences between isolates are close. In my opinion something more like <20 SNPs are close. I suggest that the authors add a reference and justification for this claim, or revise this statement.

Thank you for pointing this out for us. The genome size of *E. coli* is generally around 5 Mb, and two sequences shared less than 100 SNPs may suggest a close similarity. In a recent publication (**PMID: 32737421**), "pig isolate differed by only 119 SNPs from the human *E. coli* isolated" are also considered as genomic similarity (the last sentence in the result section in this article). **According to your grateful suggestions, we have changed our statement here.** Please see lines **254-256, 390-391.**

5) *Line 311-318 – Mostly repeat of what is already in the Results – suggest that it is revised/shortened.*

We are grateful to your suggestions. **We have rewritten this part.** Please see **lines 311-316.**

6) *Line 322-344 – Plasmid mediated carbapenem resistance was detected on farms although it is banned from usage in China. Although the authors have provided two possible explanations, a third possibility worth considering is that the plasmid and/or isolate was co-selected by use of another antimicrobial that may be present in the plasmid and/or isolate.*

Thank you for providing this useful suggestion. **We have included it in our discussion part.** Please see **lines 340-342.**

Once again. We sincerely acknowledge your contributions to the review of our manuscript. Your constructive suggestions and comments have helped us improve the presence of our work clearly.

REVIEWERS' COMMENTS

Reviewer #3 (Remarks to the Author):

I applaud the authors' patience and persistence in addressing ours and other reviewers comments. The study is once again, much improved. I have a few additional comments on the new version, most of which are minor in comparison to the last revision. Our major remaining concern is the reproducibility of the analysis and availability of data. In this regard, we believe a public GitHub repository that includes sufficient raw data (including the tree) and scripts to reproduce the key analyses and figures of the publication (where possible; we appreciate that BRIG and EasyFig figures cannot be reproduced this way), should be made available and confirmed to be functional prior to publication.

Intro

Line 43-44: This line is too a bit too sensational and the language is odd. Please change to something along the lines of "...have raised serious concerns for a potentially disastrous return to a pre-antibiotic world."

Results

Line 105: "Most of them...". 441/1871 is not most so change this to read. "The most represented provinces by isolate count were Henan (n = 191) and Hubei (n = 250), which are the two largest pig-farming provinces in China."

Line 153: '41 isolates with novel STs'. I have looked at Table S1 and notice that there are indeed 41 isolates with unidentified STs ("- " in the ST column). Without the full MLST output though, it is not possible to tell if these are truly novel alleles or not. The MLST program will output dashes (-) in the ST column for a number of reasons; sometimes it is truly a new combination of known alleles, other times it is due to novel alleles, incomplete alleles or a combination of both. That information is not present in Table S1. Rather than include it and introduce unnecessary complexity, I think it is just better to change this sentence to read "...118 distinct STs, except for 41 isolates with for which an ST could not be confidently identified."

Line 156-165: I appreciate this section is not the focus of the study but perhaps a supplementary heatmap of virulence gene carriage would be a good addition to summarise the information in Table S2.

Lines 174-177: As there is no formal statistical analysis here, I would caution the authors against using words such as 'correlation' and 'association'. It is better to say something like, "Resistance phenotypes generally corresponded to carriage of one or more genes known to confer that phenotype." Then you can also mention how many cases where phenotype was not complemented by gene carriage or vice versa. Otherwise, Figure 3 is a big improvement from the last iteration so well done.

Lines 200-201: Authors need to be careful to differentiate between where contigs or reads were used for bioinformatic analyses. Here in the results it says '..short read mapping...' but Table S5 indicates that contig files were used. Please be very clear about this in both methods and results.

Lines 211-213: Thank you for adding this analysis. I would simply change the language in this sentence to reflect that there is still some uncertainty regarding actual plasmid presence based on limitations of alignment methodology. E.g "Strikingly, other isolates that co-produced NDM and MCR (n=) displayed high sequence homology across the vast majority of these plasmid backbones, strongly suggesting carriage of highly similar plasmids."

Either add '100% identity' to all facets of the Figure S3 legend or remove them and specify that it denotes 100% identity in the figure caption.

Figures

Figure S1C: The MIC labelling here is unreadable. Please generate a legend for colours that correspond to MICs so that labelling on the figure itself is not required.

Figure 5: Please make all the data rings the same width and thinner overall so the tree topology is clearer. I am not convinced that the 'SNPs vs No of isolates' table is informative. Can probably be removed.

I also think the 'Host' metadata should be displayed on a ring instead of tips but will leave this to the author's discretion. A ring representing the major STs such as ST10, 101 and 48 could be informative too.

What is the significance of tips that are labelled with names? It is unclear.

Discussion

Line 308: Change to "...12 provinces in low proportions (3.79%)." Remove "of determination."

Line 317: should be 'tetracycline resistance genes' not "tetracycline-resistant genes"

Methods

Lines 490-495: Please specify the ST of XD35 and confirm/specify that the tree was rooted on the *E. fergusonii* sequence.

Please specify if the snippy based tree was inferred using FastTree2 or another program.

Scripts and reproducibility

We thank the authors for including their analysis script but they have not provided the raw files that the script requires to run so we cannot test it. The script and files required to run it should be uploaded to a GitHub repository where they can be freely used and rerun. See example: https://github.com/CJREID/ST58_project

*** Additional comments following provision of code and updated Figure:

- Raw data from the total collection (n=1871) that is seen in Figure 1 and Figure S1 of both sequenced and non-sequenced isolates has not been made available. Their details should be in a supplementary table; including province, source (seen in Figure 1A), AST et
- Figure 5 is nicely improved but has typos (Circiles instead of Circles)

Please provide Tables S4 and S8 as Excel not Word documents.

Reviewer #3 comments:

I applaud the authors' patience and persistence in addressing ours and other reviewers' comments. The study is once again, much improved. I have a few additional comments on the new version, most of which are minor in comparison to the last revision. Our major remaining concern is the reproducibility of the analysis and availability of data. In this regard, we believe a public GitHub repository that includes sufficient raw data (including the tree) and scripts to reproduce the key analyses and figures of the publication (where possible; we appreciate that BRIG and EasyFig figures cannot be reproduced this way), should be made available and confirmed to be functional prior to publication.

Once again, we also appreciate your patience and persistence on helping us to improve our manuscript. We have once again studied your comments below very carefully and have revised the manuscript according to these useful suggestions. We believe these suggestions not only improved our manuscript, but will also have a long-term influence on our future research work.

Also, we have revised Figure 5 according to your comments and suggestions below.

We would also like to specify that we have used an online tool iTOL (<https://itol.embl.de/>), instead of the R script, to display, annotate, and manage the trees in the manuscript in this round of modification (Lines 508-510; 834, 870). We chose this way for the tree annotation because some future readers may lack of enough experiences in bioinformatical or computer science, and it might be a bit difficult for them to repeat the data using the R script. Instead, using iTOL is an easier way; anyone who would like to repeat the data only need to submit the tree file in the website and drag/drop the annotation codes directly onto the tree. To help them to repeat the data, we have provided the tree files and the codes for annotation in supplementary materials (Data 1 for the MLST-tree in Figure 3, see lines 571-572; Data 2 for the snippy-based tree in Figure 5A, see line 574-575).

For more information, please check the help page of iTOL (<https://itol.embl.de/help.cgi>).

We believe iTOL is easier tool than the R script for readers to repeat the data.

Considering iTOL has a detailed protocol on the website (<https://itol.embl.de/help.cgi>), we do not deposit the raw data and annotation codes in gitHub.

Thank you very much.

Intro

Line 43-44: This line is too a bit too sensational and the language is odd. Please change to something along the lines of "...have raised serious concerns for a potentially disastrous return to a pre-antibiotic world."

Thank you for pointing this to us and we have removed the last part of this sentence and have changed it into "*Recently, the emergence of and rapid increases in multidrug-resistant (MDR), extensively drug-resistant (XDR), and even pandrug-resistant (PDR) bacteria, particularly bacteria resistant to last-resort drugs (carbapenems, colistin, and tigecycline), have catalyzed serious concern*". Please see lines 39-42.

Results

Line 105: "Most of them...". 441/1871 is not most so change this to read. "The most represented provinces by isolate count were Henan (n = 191) and Hubei (n = 250), which are the two largest pig-farming provinces in China."

We appreciate your good suggestion and we have changed the sentence as suggested. Please check lines 103-105 (*"The most represented provinces by isolate count were Henan (n = 191) and Hubei (n = 250), which are the two largest pig-farming provinces in China."*)

Line 153: '41 isolates with novel STs'. I have looked at Table S1 and notice that there are indeed 41 isolates with unidentified STs ("- " in the ST column). Without the full MLST output though, it is not possible to tell if these are truly novel alleles or not. The MLST program will output dashes (-) in the ST column for a number of reasons; sometimes it is truly a new combination of known alleles, other times it is due to novel alleles, incomplete alleles or a combination of both. That information is not present in Table S1. Rather than include it and introduce unnecessary complexity, I think it is just better to change this sentence to read "...118 distinct STs, except for 41 isolates with for which an ST could not be confidently identified."

This is really an excellent suggestion; we have revised the sentence accordingly. Please check lines 151-152 (*"...to 118 distinct STs, except for 41 isolates with for which an ST could not be confidently identified"*). Also, we have presented the information in Table S2 with a footnote (*"-" in columns C, D, E refers to O-types, H-types, or sequence types could not be confidently identified*).

Line 156-165: I appreciate this section is not the focus of the study but perhaps a supplementary heatmap of virulence gene carriage would be a good addition to summarise the information in Table S2.

Thank you for your suggestion. Actually, we tried to generate a heatmap to display the VFGs during our preparation of the original manuscript. However, the heatmap looks not very good enough and it is very difficult for the readers to see the genes clearly because so many genes were identified. Therefore, we finally chose to display the VFGs in an excel Table. We are sorry about this.

Lines 174-177: As there is no formal statistical analysis here, I would caution the authors against using words such as 'correlation' and 'association'. It is better to say something like, "Resistance phenotypes generally corresponded to carriage of one or more genes known to confer that phenotype." Then you can also mention how many cases where phenotype was not complemented by gene carriage or vice versa. Otherwise, Figure 3 is a big improvement from the last iteration so well done.

Thank you for pointing this out for us and also thank you for your approval on our newly generated figure 3. The statement you mentioned has been changed according to your suggestion. Please see lines 173-175 (*"Resistance phenotypes generally corresponded to carriage of one or more genes known to confer that phenotype (Figure 3)"*).

Lines 200-201: Authors need to be careful to differentiate between where contigs or reads were used for bioinformatic analyses. Here in the results it says '..short read mapping...' but Table S5 indicates that contig files were used. Please be very clear about this in both methods and results.

We agree and we have changed "short read" to "contig" in the whole manuscript. Please check line 201. Thank you.

Lines 211-213: Thank you for adding this analysis. I would simply change the language in this sentence to reflect that there is still some uncertainty regarding actual plasmid presence based on limitations of alignment methodology. E.g "Strikingly, other isolates that co-produced NDM and MCR (n=) displayed high sequence homology across the vast majority of these plasmid backbones, strongly suggesting carriage of highly similar plasmids."

Thank you for the suggestion. We have revised this place as suggested. Please see lines 213-215 (“*Strikingly, other isolates that co-produced NDM and MCR (n = 6) displayed high sequence homology across the vast majority of these plasmid backbones, strongly suggesting carriage of highly similar plasmids*”).

Either add ‘100% identity’ to all facets of the Figure S3 legend or remove them and specify that it denotes 100% identity in the figure caption.

We have added “100% identity” to all facets of the Figure S3 legend. Please check the revised Figure S3. Thank you.

Figures

Figure S1C: The MIC labelling here is unreadable. Please generate a legend for colours that correspond to MICs so that labelling on the figure itself is not required.

We have revised the figure as suggested. Please check the revised Figure S1. Thank you.

Figure 5: Please make all the data rings the same width and thinner overall so the tree topology is clearer. I am not convinced that the ‘SNPs vs No of isolates’ table is informative. Can probably be removed.

I also think the ‘Host’ metadata should be displayed on a ring instead of tips but will leave this to the author’s discretion. A ring representing the major STs such as ST10, 101 and 48 could be informative too.

We are grateful for your good suggestions. We have revised Figure 5 as suggested. First, the data rings have been adjusted to the same width and thinner. Then, the table has been removed. Third, the “Host” data has been displayed on a ring. Finally, a ring representing the major STs has been added. Please check the revised Figure 5.

What is the significance of tips that are labelled with names? It is unclear.

Thank you for your comments. We have removed the labels from the figure. Please check the revised Figure 5.

Discussion

Line 308: Change to “...12 provinces in low proportions (3.79%).” Remove “of determination.”

We have revised this sentence as suggested. Please see line 312 (“*...farms in 12 provinces in low proportions (3.79%)*”). Thank you.

Line 317: should be ‘tetracycline resistance genes’ not “tetracycline-resistant genes”

We have revised this sentence as suggested. Please see line 321 (“*...tetracycline resistance genes*”). Thank you.

Methods

Lines 490-495: Please specify the ST of XD35 and confirm/specify that the tree was rooted on the E. fergusonii sequence.

Thank you for your suggestion. The sequence type (ST746) of XD35 has been added (line 505). We have also added the statement “*the tree was rooted on the E. fergusonii sequence*” in lines 503-504.

Please specify if the snippy based tree was inferred using FastTree2 or another program.

Thank you for your suggestion. The snippy based tree was inferred using FastTree2. We have added this information (lines 504).

Scripts and reproducibility

We thank the authors for including their analysis script but they have not provided the raw files that the script requires to run so we cannot test it. The script and files required to run it should be uploaded to a GitHub repository where they can be freely used and rerun. See example: https://github.com/CJREID/ST58_project

Thank you very much for your comments. As we mentioned above in the repose of the first comment, considering some readers may lack of enough experiences in bioinformatical or computer science, and it might be a bit difficult for them to repeat the data using the R script. We therefore have used an online tool iTOL (<https://itol.embl.de/>), instead of the R script, to display, annotate, and manage the trees in the manuscript in this round of modification. It is an easier way for the tree annotation.

Considering iTOL has a detailed protocol on the website (<https://itol.embl.de/help.cgi>), we do not deposit the raw data and annotation codes in gitHub. Instead, we submit them as supplementary materials (for Figure 3, please check Data 1 in supplementary materials [lines 561-562]; for Figure 5, please check Data 2 in supplementary materials [see line 508-510]).

Additional comments following provision of code and updated Figure:

Raw data from the total collection (n=1871) that is seen in Figure 1 and Figure S1 of both sequenced and non-sequenced isolates has not been made available. Their details should be in a supplementary table; including province, source (seen in Figure 1A), AST et

We have listed this information in the newly generated Table S1. Please check it (lines 548-549).

Figure 5 is nicely improved but has typos (Circiles instead of Circles)

Sorry for the typos. We have corrected them in the figure. Please check it.

Please provide Tables S4 and S8 as Excel not Word documents.

We have redisplayed these two word files in excel files (please check the newly generated Tables S5 and S9).

Once again, we thank you very much for your time and effort on reviewing our manuscript. Hope you everything be best in this year.